# Safety Certificate against Latent Variables with Partially Unidentifiable Dynamics

**Haoming Jing** [1]   **Yorie Nakahira** [1]

## Abstract

Existing control techniques often assume access to complete dynamics or perfect simulators with fully observable states, which are necessary to verify whether the system remains within a safe set (forward invariance) or safe actions are persistently feasible at all times. However, many systems contain latent variables that make their dynamics partially unidentifiable or cause distribution shifts in the observed statistics between offline and online data, even when the underlying mechanistic dynamics are unchanged. Such "spurious" distribution shifts can break many techniques that use data to learn system models or safety certificates. To address this limitation, we propose a technique for designing probabilistic safety certificates for systems with latent variables. A key technical enabler is the formulation of invariance conditions in probability space, which can be constructed using observed statistics in the presence of distribution shifts due to latent variables. We use this invariance condition to construct a safety certificate that can be implemented efficiently in real-time control. The proposed safety certificate can persistently find feasible actions that control long-term risk to stay within tolerance. Stochastic safe control and (causal) reinforcement learning have been studied in isolation until now. To the best of our knowledge, the proposed work is the first to use causal reinforcement learning to quantify long-term risk for the design of safety certificates. This integration enables safety certificates to efficiently ensure long-term safety in the presence of latent variables. The effectiveness of the proposed safety certificate is demonstrated in numerical simulations.

[1]Electrical and Computer Engineering Department, Carnegie Mellon University, Pittsburgh, USA. Correspondence to: Yorie Nakahira <ynakahir@andrew.cmu.edu>.

*Proceedings of the 42nd International Conference on Machine Learning*, Vancouver, Canada. PMLR 267, 2025. Copyright 2025 by the author(s).

## 1. Introduction

Autonomous control systems must operate safely even in the presence of latent variables. For instance, autonomous ground vehicles must anticipate objects suddenly emerging from behind occlusions or pedestrians unexpectedly changing their intent to cross the road. In such scenarios, risk-critical variables may be unobservable. These *latent* variables can induce distribution shifts between offline and online settings in visible variables and render the partial models associated with the latent variables unidentifiable. Therefore, in the design of control policies, one must explicitly differentiate between statistical models (Schölkopf et al., 2021; Ye et al., 2024), which capture correlations in observed states, and mechanistic models, which encode the causal structure of the system. Designing safety certificates directly from data without this distinction can lead to vulnerabilities from spurious distribution shifts–divergence between offline versus online statistics even when the underlying dynamics remain unchanged (see Section 2.2 for mathematical details). Moreover, the effects of latent variables on observed states may remain subtle within a short time horizon, but by the time they become apparent, corrective action may no longer be feasible. For example, if a child suddenly emerges from behind a large bus, the vehicle may not have sufficient time to stop safely. The presence of such irrecoverable states further complicates safety assurance. While myopic safety can often be efficiently enforced, it is insufficient for ensuring long-term safety. However, the complexity of safety assurance increases unfavorably with the time horizon, particularly in the presence of latent variables.

Motivated by these challenges, this paper explores the following research question:

*How can we efficiently assure long-term safety for stochastic systems in the presence of latent variables, which induce distribution shifts in offline vs. online statistics and partially unobservable dynamics?*

Existing safety certificates often rely on complete system models or fully observable states to verify whether the system remains within a safe set and whether safe actions exist continuously (Hsu et al., 2023; Wabersich et al., 2023).

These methods commonly determine actions that satisfy forward invariance conditions in the state space (Blanchini, 1999), which require full knowledge of system dynamics and state observability for evaluation. However, many real-world systems do not meet these requirements due to the presence of latent variables. While myopic controllers are more practical for real-time control with limited onboard resources (Ames et al., 2016; 2019), they may fail to guarantee long-term safety due to the presence of irrecoverable states and the accumulation of tail-risk events over time. On the other hand, achieving long-term safety—especially in systems with latent variables—often demands complex approaches to handle distribution shifts (e.g., reevaluation of long-term probabilities or retraining using new data), which impose additional challenges in real-time control.

In this paper, we focus on safety certificates for stochastic systems with latent variables, uncertainties with unbounded support, and actuation limits. We formulate invariance conditions in probability space in a way that accounts for latent variables and distribution shifts and can be computed using observed statistics. First, we show a relation between risk measures and marginalized value or Q-functions by employing a modified Bellman equation—adapted to account for latent variables. Next, we present the conditions on the control action that are sufficient to manage risk within a specified tolerance, based on invariance conditions in probability space. Based on this relation, we then show action constraints can be obtained from observed statistics, even in the presence of distribution shifts between observed offline vs. online statistics. In particular, action constraints are constructed to assure long-term safety with persistent feasibility, and this construction leverages probabilistic invariance and the inherent conditions satisfied by marginalized value or Q-functions. These conditions are then utilized to design a safety certificate that can be used by a myopic controller to guarantee long-term safety. This approach also allows the design of safety certificates to easily exploit the large existing body of literature from such domains: the risk measure can be evaluated using existing risk quantification methods, and its equivalent form in marginalized value function and Q-function can be evaluated by causal reinforcement learning technique.

## 2. Problem Statement

### 2.1. System Model

We consider a confounded Markov decision process described by the tuple $(\mathcal{X}, \mathcal{U}, \mathcal{W}, \mathcal{P}, H)$[1]. Here, $X_t \in \mathcal{X}$ is the observable (visible) state, $U_t \in \mathcal{U}$ is the action, $W_t \in \mathcal{W}$

is the unobservable latent variable, and $H \in \mathbb{Z}_+$ is the length of the episode. The mechanistic model is given by $\mathcal{P}(X_{t+1}, W_{t+1}|X_t, W_t, U_t)$, which captures the transition of the underlying system dynamics.

We make the following assumption about the latent variable. Here, the notation $\perp$ denotes statistical independence.

**Assumption 2.1.** The sequence $\{W_t\}$ satisfies

$$W_t \perp \{W_\tau\}_{\tau < t}, \{X_\tau\}_{\tau < t}, \{U_\tau\}_{\tau \le t} | X_t. \qquad (1)$$

Due to Assumption 2.1, the transition kernel admits the decomposition $\mathcal{P}(X_{t+1}, W_{t+1}|X_t, W_t, U_t) = \mathcal{P}(W_t|X_t)\mathcal{P}(X_{t+1}|X_t, U_t, W_t)$. This condition gives Markovian property in the observable state $X_t$, i.e., [2]

$$P(X_{t+1}|X_t, U_t) = P(X_{t+1}|\{X_\tau\}_{\tau \le t}, \{U_\tau\}_{\tau \le t}). \qquad (2)$$

Let $\mathcal{D} = \{\mathcal{D}_1, \mathcal{D}_2, \cdots, \mathcal{D}_{N_D}\}$ denote the available offline data. Here, $N_D$ is the size of the training dataset, and each individual data $\mathcal{D}_i, i \in \{1, 2, \cdots, N_D\}$ contains the sequence of observable state $\{X_t\}_{t \in \{0,1,\cdots,H\}} := X_{0:H}$ and control action $U_{0:H}$ in an episode. The control actions are generated by a behavioral policy $\pi^b$, i.e., $U_t \sim \pi^b(U_t|X_t, W_t)$, which is assumed to be unknown. Accordingly, in dataset $\mathcal{D}$, the observable state satisfies the following offline statistics:

$$P_{\text{offline}}(X_{t+1}|X_t, U_t) = $$
$$\frac{\mathbb{E}_{W_t \sim \mathcal{P}(W_t|X_t)}[\mathcal{P}(X_{t+1}|X_t, U_t, W_t)\pi^b(U_t|X_t, W_t)]}{\mathbb{E}_{W_t \sim \mathcal{P}(W_t|X_t)}[\pi^b(U_t|X_t, W_t)]}. $$
$$(3)$$

On the other hand, the online statistics of the observable state $X_t$ is given by

$$P_{\text{online}}(X_{t+1}|X_t, U_t) := $$
$$\mathbb{E}_{W_t \sim \mathcal{P}(W_t|X_t)}[\mathcal{P}(X_{t+1}|X_t, U_t, W_t)], \qquad (4)$$

where the latent variable $W_t$ is marginalized. The unobservable nature of the latent variable also causes a mismatch between the online statistics (4) and the offline statistics (3) of the offline data $\mathcal{D}$.

Unlike the behavioral policy from the offline setting, in the online setting, a decision policy cannot depend on the latent variable because the latent variable $W$ cannot be observed by an online controller. The online policy is designed to

---

[1]To avoid confusion with later context, we do not include reward when describing the process. It is a trivial extension to incorporate reward and maximize the reward during control by adding the reward to the optimization objective in (55).

[2]Throughout the paper, we use $\mathcal{P}$ to denote the mechanistic model, which is the transition kernel including the complete state (observable and latent) in a Markov decision process or a confounded Markov decision process. On the other hand, we use $P$ to denote the statistical model, which is either the offline statistics (e.g., (3)) or the online statistics (e.g., (4)) that only involves the observable states.

satisfy multifaceted design considerations. Some objectives, such as performance objectives, are captured in a nominal policy $\pi^n$, *i.e.,* $U_t \sim \pi^n(\cdot|X_t)$. The safety objective is ensured by a safety certificate. The safety certificate is represented by a mapping $S : \mathcal{X} \times \mathcal{U} \times \mathbb{Z} \to \mathbb{R}$, where a control action $U$ is considered to be safe with respect to a state $X$ at time $t$ when the constraint $S(X_t, U_t, t) \geq 0$ is not violated. Here, safety is characterized by an event $C(X_t)$ that can be evaluated by the state $X_t$. For example, a common definition for safety is that the state must remain in a safe set $\mathcal{C}$. In this example, we have $C(X_t) = \{X_t \in \mathcal{C}\}$. The long-term safety with respect to a certain control policy $\pi$ at time $t$ is quantified by the long-term safe probability

$$\mathbb{P}^\pi(C(X_t) \cap C(X_{t+1}) \cap \cdots \cap C(X_H)), \qquad (5)$$

where the probability is calculated assuming the use of policy $\pi$ in the closed loop system with the online statistics $P_{\text{online}}$. Here, the policy $\pi$ has the form $\pi : \mathcal{X} \times \mathbb{Z} \to \Delta(\mathcal{U})$, *i.e.,* a mapping from $x \in \mathcal{X}$ and time $t \in \mathbb{Z}$ to a distribution of control action in the action space $\mathcal{U}$, which we denote as $\Delta(\mathcal{U})$[3].

In this paper, our goal is to study the design technique for safety certificates that ensure not only that the safety of the immediate future state but also that states irrecoverably leading to risk (no feasible control action leads the system to safe future states) are avoided. Specifically, the proposed technique assures that the long-term safe probability conditioned on the initial observable state $X_0$ is not smaller than a threshold $1 - \epsilon$ for the entire episode, *i.e.,*

$$\mathbb{P}^{\hat{\pi}, \pi}(C(X_t) \cap C(X_{t+1}) \cap \cdots \cap C(X_H)|X_0) \geq 1 - \epsilon,$$
$$\forall t \in \{0, 1, \cdots, H\}, \qquad (6)$$

given certain initial conditions. Here, the probability is calculated assuming the use of online policy $\hat{\pi}$ for times $\{0, 1, \cdots, t-1\}$ and policy $\pi$ for times $\{t, t+1, \cdots, H-1\}$ in the closed loop system with the online statistics $P_{\text{online}}$. This is achieved by characterizing the action constraints $S(X_t, U_t, t) \geq 0$ that are sufficient to assure the safety objective (6) and are continuously feasible at all time $t$.

### 2.2. Technical Challenges

Due to the presence of latent variables, samples of *complete* state $\{X_t, W_t\}_t$ cannot be obtained, and thus the underlying transition dynamics $\mathcal{P}(X_{t+1}, W_{t+1}|X_t, W_t, U_t)$ is not fully identifiable. On the other hand, one cannot treat the observed statistics (statistics of the observed variable $\{X_t\}_t$) as if it is the mechanistic model (underlying transition dynamics). This is because, even when the mechanistic

model $\mathcal{P}(X_{t+1}, W_{t+1}|X_t, W_t, U_t)$ remains unchanged, the observed statistics can have distribution shifts between the offline vs. online settings: *i.e.,* $P_{\text{online}}(X_{t+1}|X_t, U_t) \neq P_{\text{offline}}(X_{t+1}|X_t, U_t)$. Such "spurious" distribution shifts can fail many data-driven techniques that do not explicitly differentiate observed statistics vs. mechanistic models, and prohibit the use of existing stochastic safe control techniques that require accurate transition dynamics or perfect simulators.[4] Similarly, due to such distribution shifts, one may obtain misleading estimates of risk probability and probabilistic reachability when they are learned solely from data (observed offline statistics).[5] However, this vulnerability has received little attention in stochastic safe control.

### 2.3. Related Work

#### 2.3.1. SAFE CONTROL

Many design techniques are developed for the safety certificate in stochastic or deterministic dynamical systems. These techniques include barrier/Lyapunov functions (Nishimura & Hoshino, 2024; Wang et al., 2021a; Vahs et al., 2023; Jahanshahi et al., 2020; Dean et al., 2021), barrier certificates (Prajna et al., 2007; Ahmadi et al., 2020; 2018), and predictive safety filters (Wabersich & Zeilinger, 2018; 2023; 2021; Wabersich et al., 2021). For partially observed systems with known dynamics, existing literature has proposed robust control barrier functions for deterministic systems with bounded estimation errors (Zhang et al., 2022; Dean et al., 2021; Wang & Xu, 2022; Qin et al., 2022b; Zhao & Yu, 2024), as well as control barrier functions and barrier certificates constructed on belief space or estimated state for stochastic systems (Vahs et al., 2023; Ahmadi et al., 2020; Carr et al., 2023; Jahanshahi et al., 2022; Clark, 2019; Dean et al., 2021; Jahanshahi et al., 2020; Ahmadi et al., 2018). When perfect simulators are available, barrier/Lyapunov functions can be designed through optimization problems that check the existence of functions satisfying barrier/Lyapunov function conditions at all states using sampled data (Nejati & Zamani, 2023; Anand & Zamani, 2023; Dai et al., 2023; Qin et al., 2022a; Wang et al., 2023; Lindemann et al., 2021; Xiao et al., 2023). These techniques are commonly built based forward invariance conditions in the state space[6] (Blanchini, 1999). Some techniques also construct safety conditions based on invariance conditions in the probability space (Wang et al., 2022; Jing & Nakahira, 2022) or its low dimensional representation (Hoshino et al., 2023) to ensure long-term safety. These conditions commonly *require complete transition dynamics and fully ob-*

---

[3]In Section 3, we define the augmented state $\hat{Y}$ which captures time, so that the input to the policy $\pi$ is $\hat{Y}$.

[4]For example, Figure 1 and 2 show that the safety guarantee from these techniques can fail under such a distribution shift.

[5]See Appendix A for an example

[6]The system state stays within a certain set if it originated from within the set. The information of the full system state and state dynamics is needed to check this condition.

*servable states to evaluate.* It is also difficult to check such conditions using noisy data in stochastic systems or biased offline data resulting from the presence of latent variables.

Existing work has also studied the techniques to handle distribution shifts. A common approach is to avoid distribution shifts by constraining the system to stay within the states with known distribution. For example, Lyapunov density model is used to constrain the state within the distribution sampled in the offline training data (Kang et al., 2022; Castaneda et al., 2023; Wu et al., 2023). Another approach assumes all possible transition distributions are known or can be samples, or policies or safety certificates for all possible distributions can be sampled in advance. For instance, meta-learning is used to learn effective policies under different distributions (Guan et al., 2024; Richards et al., 2023). Our problems differ from the first approach in the sense that the distribution shifts (arising from latent variables) cannot be avoided. Our problem differ from the second approach in the sense that ours do not have access to offline samples for all possible statistics of observed variables, and data with latent variables are not accessible. In addition, these approaches focus on addressing distribution shifts caused by changes in the system dynamics, while our approach considers distribution shifts purely due to the presence of latent variables even if the system dynamics is unchanged. Little safe control literature handles this type of distribution shift, and, to the best of our knowledge, this paper is the first one to do so.

### 2.3.2. CAUSAL REINFORCEMENT LEARNING

There has been extensive work in causal reinforcement learning that aim to address the biasing (confounding) effect of the latent variables. Existing works have developed estimation methods for value function and/or Q-function in the context of confounded Markov decision process (Wang et al., 2021b; Chen et al., 2022; Bennett et al., 2021; Shi et al., 2024; Fu et al., 2022; Xu et al., 2023) and partially observable Markov decision process (Bennett & Kallus, 2024; Miao et al., 2022; Shi et al., 2022). Many algorithms are developed for settings with different available information, such as the availability of backdoor/frontdoor adjustment variable (Wang et al., 2021b), proxy variables (Miao et al., 2022; Bennett & Kallus, 2024), and instrumental variables (Chen et al., 2022; Fu et al., 2022). While these approaches offer techniques to manage latent variables in diverse settings, to the best of our knowledge, *none have been applied to stochastic safe control problems with persistent feasibility guarantee.* This paper bridges that gap by introducing a framework that integrates causal reinforcement learning into safety certificate design. Although there exist methods such as Srinivasan et al. 2020 that use Q-function to represent safety and integrate safe control with learning, critical assumptions about the existence of actions

that brings the state to safety has to be made. Our method does not rely on such assumptions and gives persistent feasibility guarantee on the control policy. While we employ the method of Shi et al. 2024 for safe controller design, the proposed framework is expected to generalize to other causal reinforcement learning techniques and their respective settings.

## 3. Proposed Method

Before introducing the proposed method, we first define a function $\Psi^\pi : \mathcal{X} \times \mathbb{Z} \to [0, 1]$ that captures the long-term safety probability with respect to a policy $\pi$ conditioned on a state $x$:

$$\Psi^\pi(x, t) :=$$
$$\mathbb{P}^\pi(C(X_t) \cap C(X_{t+1}) \cap \cdots \cap C(X_H)|X_t = x), \quad (7)$$

where $t \leq H$. Here, the probability is evaluated with the assumption that the sequence $X_{t:H}$ has statistics (4). We then define two auxiliary Markov decision processes (MDPs). The first MDP is described by the tuple $(\mathcal{Y}, \mathcal{U}, \tilde{\mathcal{P}}_{\text{online}}, r, H)$, where $\hat{Y}_t := [\hat{X}_t^T, K_t]^T \in \mathcal{X} \times \mathbb{Z} := \mathcal{Y}$ is the state, and $U_t \in \mathcal{U}$ is the control action. In this process, the sequence $\hat{X}_{0:H}$ has the online statistics (4) when $C(\hat{X}_t)$ occurs and the transition $\hat{X}_{t+1} = \hat{X}_t$ when $C(\hat{X}_t)$ does not occur, *i.e.,*[7]

$$\tilde{\mathcal{P}}_{\text{online}}(\hat{X}_{t+1}|\hat{X}_t, U_t)$$
$$= \begin{cases} P_{\text{online}}(X_{t+1}|X_t, U_t), & C(\hat{X}_t) \text{ occurs} \\ \delta(\hat{X}_{t+1} - \hat{X}_t), & C(\hat{X}_t) \text{ doesn't occur,} \end{cases} \quad (8)$$

where $\delta(\cdot)$ is the Dirac delta function. We define $K_{0:H}$ to be a sequence that captures the remaining time in the episode. Its statistics satisfies $K_{t+1} = K_t - 1$, *i.e.,*

$$\tilde{\mathcal{P}}_{\text{online}}(K_{t+1}|K_t) = \delta(K_{t+1} - K_t + 1). \quad (9)$$

The transition kernel of the MDP is given by $\tilde{\mathcal{P}}_{\text{online}}(\hat{Y}_{t+1}|\hat{Y}_t, U_t) = \tilde{\mathcal{P}}_{\text{online}}(\hat{X}_{t+1}|\hat{X}_t, U_t)\tilde{\mathcal{P}}_{\text{online}}(K_{t+1}|K_t)$. The reward function $r : \mathcal{Y} \to \{0, 1\}$ is defined by

$$r([x^T, k]^T) = \mathbf{1}\{k = 0\}\mathbf{1}\{C(x)\}, \quad (10)$$

where $\mathbf{1}\{\mathcal{E}\}$ is the indicator function which takes the value 1 if event $\mathcal{E}$ occurs and 0 otherwise. The second MDP is described by the tuple $(\mathcal{Y}, \mathcal{U}, \tilde{\mathcal{P}}_{\text{offline}}, r, H)$, where $\bar{Y}_t := [\bar{X}_t^T, K_t]^T \in \mathcal{Y}$ is the state. In this process, the sequence $\bar{X}_{0:H}$ has the offline statistics (3) when $C(\bar{X}_t)$ occurs and

---

[7]Here, with slight abuse of notation, we use $\tilde{\mathcal{P}}_{\text{online}}(\hat{X}_{t+1}|\hat{X}_t, U_t) = P_{\text{online}}(X_{t+1}|X_t, U_t)$ to represent that, when $\hat{X}_t = X_t$, $\hat{X}_{t+1}$ has the same distribution as $X_{t+1}$ when the statistics of $X_{0:H}$ is $P_{\text{online}}$. We use this notation system in (11) as well.

the transition $\bar{X}_{t+1} = \bar{X}_t$ when $C(\bar{X}_t)$ does not occur, *i.e.,*

$$
\begin{aligned}
&\tilde{\mathcal{P}}_{\text{offline}}(\bar{X}_{t+1}|\bar{X}_t, U_t) \\
&= \begin{cases} P_{\text{offline}}(X_{t+1}|X_t, U_t), & C(\bar{X}_t) \text{ occurs} \\ \delta(\bar{X}_{t+1} - \bar{X}_t), & C(\bar{X}_t) \text{ doesn't occur.} \end{cases}
\end{aligned}
\tag{11}
$$

The transition kernel of the MDP is given by $\tilde{\mathcal{P}}_{\text{offline}}(\bar{Y}_{t+1}|\bar{Y}_t, U_t) = \tilde{\mathcal{P}}_{\text{offline}}(\bar{X}_{t+1}|\bar{X}_t, U_t)\tilde{\mathcal{P}}_{\text{offline}}(K_{t+1}|K_t)$, where $\tilde{\mathcal{P}}_{\text{offline}}(K_{t+1}|K_t) = \tilde{\mathcal{P}}_{\text{online}}(K_{t+1}|K_t)$.

The rest of Section 3 is organized as follows. In section 3.1, we show that certain value function defined for the MDP $(\mathcal{Y}, \mathcal{U}, \tilde{\mathcal{P}}_{\text{online}}, r, H)$ is equal to the function $\Psi^\pi$. In Section 3.2, we introduce a safety certificate formulated based on the value function and show that the satisfaction of the safety certificate provably ensures the safety objective (6). In Section 3.3, we propose an equivalent condition to the safety certificate that can be evaluated using certain Q-function defined for the MDP $(\mathcal{Y}, \mathcal{U}, \tilde{\mathcal{P}}_{\text{online}}, r, H)$. We then show that there always exists a control action $U_t \in \mathcal{U}$ such that this condition is satisfied. In Section 3.4, we show that, using offline dataset $\mathcal{D}$, one can obtain offline dataset $\tilde{\mathcal{D}}$ that has sequences with statistics $\tilde{\mathcal{P}}_{\text{offline}}$, which can be used to learn value function and/or Q-function defined for the MDP $(\mathcal{Y}, \mathcal{U}, \tilde{\mathcal{P}}_{\text{online}}, r, H)$ with existing causal reinforcement learning methods. We then propose an integrated safe control algorithm using one existing causal reinforcement learning method as an example.

### 3.1. Value Function Representation for Long-term Safe Probability

We consider the value function representation inspired by (Hoshino & Nakahira, 2024). We define the marginalized value function $V : \mathcal{Y} \to [0, 1]$ and the marginalized Q-function $Q : \mathcal{Y} \times \mathcal{U} \to [0, 1]$ with respect to the MDP $(\mathcal{Y}, \mathcal{U}, \tilde{\mathcal{P}}_{\text{online}}, r, H)$:

$$
V^\pi([x^T, k]^T) := \mathbb{E}_{\tilde{\mathcal{P}}_{\text{online}}}[\sum_{\tau=0}^{k} r(\hat{Y}_\tau) | \hat{Y}_0 = [x^T, k]^T, \pi]
\tag{12}
$$

$$
= \mathbb{E}_{\tilde{\mathcal{P}}_{\text{online}}}[\sum_{\tau=0}^{\infty} r(\hat{Y}_\tau) | \hat{Y}_0 = [x^T, k]^T, \pi]
\tag{13}
$$

$$
Q^\pi([x^T, k]^T, u) :=
$$
$$
\mathbb{E}_{\tilde{\mathcal{P}}_{\text{online}}}[\sum_{\tau=0}^{k} r(\hat{Y}_\tau) | \hat{Y}_0 = [x^T, k]^T, U_0 = u, \pi]
\tag{14}
$$

$$
= \mathbb{E}_{\tilde{\mathcal{P}}_{\text{online}}}[\sum_{\tau=0}^{\infty} r(\hat{Y}_\tau) | \hat{Y}_0 = [x^T, k]^T, U_0 = u, \pi].
\tag{15}
$$

Here, we may sum to infinity because, given $\hat{Y}_0 = [x^T, k]^T$, we have $r(\hat{Y}_\tau) = 0$ for all $\tau > k$ due to definition (10).

Throughout this paper, we use the subscript in the expectation to denote the distribution or the transition kernel where the expectation is taken over.

**Proposition 3.1.** *Consider the marginalized value function defined in* (12) *for* $\tilde{\mathcal{P}}_{online}$ *and the long-term safe probability defined in* (7) *for* $P_{online}$*. We have*

$$
V^\pi([x^T, k]^T) = \Psi^\pi(x, H - k)
\tag{16}
$$

*for all* $x \in \mathcal{X}$ *and* $k \in \mathbb{Z}$*.*

The proof is given in Appendix B.

### 3.2. Safety Condition

Here, we present a sufficient condition to satisfy the safety objective (6) using the value function representation. We consider the condition

$$
\mathbb{E}_{\tilde{\mathcal{P}}_{\text{online}}(\hat{Y}_{t+1}|\hat{Y}_t, U_t)}[V^\pi(\hat{Y}_{t+1})|\hat{Y}_t, U_t] - V^\pi(\hat{Y}_t) \geq 0, \tag{17}
$$

where $\hat{Y}_t = [X_t^T, H - t]^T, \forall t \in \{0, 1, \cdots, H - 1\}$.

**Theorem 3.2.** *Consider the marginalized value function defined in* (12) *for* $\tilde{\mathcal{P}}_{online}$ *and the long-term safe probability defined in* (7) *for* $P_{online}$*. If* $\Psi^\pi(X_0, 0) > 1 - \epsilon$ *and the condition* (17) *is satisfied at all times* $t \in \{0, 1, \cdots, H-1\}$*, then the safety objective* (6) *for the system with transition kernel* $\mathcal{P}$ *and online statistics* $P_{online}$ *holds.*

*Proof.* We have

$$
\begin{aligned}
&\mathbb{P}^{\hat{\pi}, \pi}(C(X_t) \cap C(X_{t+1}) \cap \cdots \cap C(X_H)|X_0) \\
&= \mathbb{E}_{P_{\hat{\pi}}(X_t|X_0)}[\Psi^\pi(X_t, t)|X_0],
\end{aligned}
\tag{18}
$$

where $P_{\hat{\pi}}(X_t|X_0)$ is the conditional distribution of $X_t$ given $X_0$ assuming the sequence $X_{0:t}$ has statistics $P_{online}$ and a policy $\hat{\pi}$ is used for times $\{0, 1, \cdots, t-1\}$. From Proposition 3.1, we have that $V^\pi([x^T, k]^T) = \Psi^\pi(x, H - k)$. Therefore, to prove the theorem, it suffices to prove the following statement: if

$$
V^\pi([X_0^T, H]) := V^\pi(\hat{Y}_0) > 1 - \epsilon,
\tag{19}
$$

and the condition (17) is satisfied at all times $t \in \{0, 1, \cdots, H - 1\}$, then

$$
\mathbb{E}_{\tilde{P}_{\hat{\pi}}(\hat{Y}_t|\hat{Y}_0)}[V^\pi(\hat{Y}_t)|\hat{Y}_0] \geq 1 - \epsilon
\tag{20}
$$

holds for all $t \in \{0, 1, \cdots, H\}$, where $\tilde{P}_{\hat{\pi}}(\hat{Y}_t|\hat{Y}_0)$ is the conditional distribution of $\hat{Y}_t$ given $\hat{Y}_0$ assuming the sequence $\hat{Y}_{0,t}$ has statistics $\tilde{\mathcal{P}}_{\text{online}}$ and a policy $\hat{\pi}$ is used for times $\{0, 1, \cdots, t-1\}$. Note that we consider the policy $\hat{\pi}$ to be the online policy here. Since the online policy produces a deterministic control action, we define the online policy as a mapping $\hat{\pi} : \mathcal{Y} \to \mathcal{U}$ so that we have $\hat{\pi}(\hat{Y}_t) = U_t$ for all

$t \in \{0, 1, \cdots, H - 1\}$. We show (20) holds for all times $t \in \{0, 1, \cdots, H\}$ using mathematical induction. We first show that (20) holds at time 0. We have

$$V^\pi(\hat{Y}_0) = \mathbb{E}_{\tilde{P}_{\hat{\pi}}(\hat{Y}_0|\hat{Y}_0)}[V^\pi(\hat{Y}_0)|\hat{Y}_0] \geq 1 - \epsilon \qquad (21)$$

holds due to (19). We then show that, given (20) holds at time $t$, it also holds at time $t + 1$. Taking conditional expectation over the conditional distribution $\tilde{P}_{\hat{\pi}}(\hat{Y}_0|\hat{Y}_0)$ on both side of (17) yields

$$\mathbb{E}_{\tilde{P}_{\hat{\pi}}(\hat{Y}_t|\hat{Y}_0)}[\mathbb{E}_{\tilde{\mathcal{P}}_{\text{online}}(\hat{Y}_{t+1}|\hat{Y}_t, U_t)}[V^\pi(\hat{Y}_{t+1})|\hat{Y}_t, U_t]|\hat{Y}_0]$$
$$\geq \mathbb{E}_{\tilde{P}_{\hat{\pi}}(\hat{Y}_t|\hat{Y}_0)}[V^\pi(\hat{Y}_t)|\hat{Y}_0]. \qquad (22)$$

From the law of total expectation, we have

$$\mathbb{E}_{\tilde{P}_{\hat{\pi}}(\hat{Y}_t|\hat{Y}_0)}[\mathbb{E}_{\tilde{\mathcal{P}}_{\text{online}}(\hat{Y}_{t+1}|\hat{Y}_t, U_t)}[V^\pi(\hat{Y}_{t+1})|\hat{Y}_t, U_t]|\hat{Y}_0]$$
$$= \mathbb{E}_{\tilde{P}_{\hat{\pi}}(\hat{Y}_t|\hat{Y}_0)}[\mathbb{E}_{\tilde{\mathcal{P}}_{\text{online}}(\hat{Y}_{t+1}|\hat{Y}_t, \hat{\pi}(\hat{Y}_t))}[V^\pi(\hat{Y}_{t+1})|\hat{Y}_t, \hat{\pi}(\hat{Y}_t)]|\hat{Y}_0]$$
$$(23)$$
$$= \mathbb{E}_{\tilde{P}_{\hat{\pi}}(\hat{Y}_{t+1}|\hat{Y}_0)}[V^\pi(\hat{Y}_{t+1})|\hat{Y}_0]. \qquad (24)$$

Therefore, we have

$$\mathbb{E}_{\tilde{P}_{\hat{\pi}}(\hat{Y}_{t+1}|\hat{Y}_0)}[V^\pi(\hat{Y}_{t+1})|\hat{Y}_0]$$
$$\geq \mathbb{E}_{\tilde{P}_{\hat{\pi}}(\hat{Y}_t|\hat{Y}_0)}[V^\pi(\hat{Y}_t)|\hat{Y}_0] \qquad (25)$$
$$\geq 1 - \epsilon, \qquad (26)$$

which gives that (20) holds at time $t + 1$. $\qquad\square$

### 3.3. Evaluation of Safety Condition and Persistent Feasibility Guarantee

Evaluating (17) can be difficult, since even if the marginalized optimal value function $V^\pi$ is available in closed form, the term $\mathbb{E}_{\tilde{\mathcal{P}}_{\text{online}}(\hat{Y}_{t+1}|\hat{Y}_t, U_t)}[V^\pi(\hat{Y}_{t+1})|\hat{Y}_t, U_t]$ cannot be evaluated since the distribution $\tilde{\mathcal{P}}_{\text{online}}(\hat{Y}_{t+1}|\hat{Y}_t, U_t)$ is unknown. Here, we show a condition that guarantees the satisfaction of (17) and can be evaluated with only the marginalized Q-function $Q^\pi$:

$$S(X_t, U_t, t) :=$$
$$Q^\pi(\hat{Y}_t, U_t) - \mathbb{E}_{U \sim \pi(U|\hat{Y}_t)}[Q^\pi(\hat{Y}_t, U)|\hat{Y}_t] \geq 0. \qquad (27)$$

where $\hat{Y}_t = [X_t^T, H - t]^T, \forall t \in \{0, 1, \cdots, H - 1\}$. This formulation allows the Q-function obtained from causal reinforcement learning techniques to be used for evaluating the safety condition. We then show that the satisfaction of this safety condition implies the satisfaction of the safety condition (17).

**Lemma 3.3.** *Consider the marginalized value function defined in (12) and the marginalized Q-function defined in (14). For all times $t \in \{0, 1, \cdots, H - 1\}$, if the control action $U_t \in \mathcal{U}$ satisfies (27), then it also satisfies (17).*

*Proof.* For all times $t \in \{0, 1, \cdots, H - 1\}$, the modified Bellman equation gives

$$Q^\pi(\hat{Y}_t, U_t) = r(\hat{Y}_t) + \mathbb{E}[V^\pi(\hat{Y}_{t+1})|\hat{Y}_t, U_t] \qquad (28)$$
$$= \mathbb{E}[V^\pi(\hat{Y}_{t+1})|\hat{Y}_t, U_t] \qquad (29)$$

since $r(\hat{Y}_t) = 0$ for all times $t \neq H$. By definitions (12) and (14), we have

$$V^\pi(\hat{Y}_t) = \mathbb{E}_{u \sim \pi(U|\hat{Y}_t)}[Q^\pi(\hat{Y}_t, U)|\hat{Y}_t]. \qquad (30)$$

Combining (29) and (30), we have

$$S(X_t, U_t, t) = Q^\pi(\hat{Y}_t, U_t)$$
$$- \mathbb{E}_{U \sim \pi(U|\hat{Y}_t)}[Q^\pi(\hat{Y}_t, U)|\hat{Y}_t] \qquad (31)$$
$$= \mathbb{E}[V^\pi(\hat{Y}_{t+1})|\hat{Y}_t, U_t] - V^\pi(\hat{Y}_t). \qquad (32)$$
$$\square$$

To ensure the safety objective, there must always exist feasible control action that doesn't violate the safety condition (27). Here, we present a provable guarantee for such persistent feasibility.

**Theorem 3.4.** *For all times $t \in \{0, 1, \cdots, H - 1\}$, there always exists $U_t \in \mathcal{U}$ such that (27) holds.*

*Proof.* Consider

$$u^* = \arg\max_{u \in \mathcal{U}} Q^\pi(\hat{Y}_t, u), \qquad (33)$$

we have

$$Q^\pi(\hat{Y}_t, u^*) \geq Q^\pi(\hat{Y}_t, u), \forall u \in \mathcal{U}. \qquad (34)$$

We also have

$$\int_{u \in \mathcal{U}} \pi(U = u|\hat{Y}_t = y)du = 1, \forall y \in \mathcal{Y}. \qquad (35)$$

Due to (34) and (35), we have

$$Q^\pi(\hat{Y}_t, u^*) = Q^\pi(\hat{Y}_t, u^*) \int_{u \in \mathcal{U}} \pi(U = u|\hat{Y}_t)du \qquad (36)$$
$$= \int_{u \in \mathcal{U}} Q^\pi(\hat{Y}_t, u^*)\pi(U = u|\hat{Y}_t)du \qquad (37)$$
$$\geq \int_{u \in \mathcal{U}} Q^\pi(\hat{Y}_t, u)\pi(U = u|\hat{Y}_t)du \qquad (38)$$
$$= \mathbb{E}_{U \sim \pi(U|\hat{Y}_t)}[Q^\pi(\hat{Y}_t, U)|\hat{Y}_t]. \qquad (39)$$

As $U_t = u^* \in \mathcal{U}$ satisfies (27), there exists a control action in $\mathcal{U}$ that satisfies (27). $\qquad\square$

### 3.4. Proposed Algorithm

Before introducing the proposed algorithm, we first show that, even if the MDP $(\mathcal{Y}, \mathcal{U}, \tilde{\mathcal{P}}_{\text{offline}}, r, H)$ is an auxiliary process and does not have the corresponding physical system, we can obtain dataset $\tilde{\mathcal{D}} = \{\tilde{\mathcal{D}}_1, \tilde{\mathcal{D}}_2, \cdots, \tilde{\mathcal{D}}_{N_D}\}$ using the available dataset $\mathcal{D}$. Here, each individual data $\tilde{\mathcal{D}}_i$ contains the sequence of state $\hat{Y}_{0:H}^i$ and control action $U_{0:H}^i$ in an episode, and the sequences follow the statistics $\tilde{P}_{\text{offline}}$. We propose Algorithm 1 that generates $\tilde{\mathcal{D}}$ using $\mathcal{D}$. Using data $\tilde{\mathcal{D}}$, one can estimate the value function and Q-function defined in (12) and (14) using existing causal reinforcement learning methods. Here, we introduce an example method that uses the mediator variable to learn unbiased Q-function (Shi et al., 2024). This method utilizes the front-door adjustment (Pearl, 2009) to counter the confounding effect. Note that the model and assumption introduced in this subsection are *specific to the application of the corresponding method only*. The proposed method works with any causal safe control methods whose model satisfies Assumption 2.1.

---

**Algorithm 1** Generation of $\tilde{\mathcal{D}}$ using $\mathcal{D}$

---

1: **Input:** offline dataset $\mathcal{D}$
2: $\tilde{\mathcal{D}} \leftarrow \emptyset$
3: **for** $i$ in $\{1, 2, \cdots, N_D\}$ **do**
4: $\quad \{X_{0:H}^i, U_{0:H}^i\} \leftarrow \mathcal{D}_i$
5: $\quad \hat{X}_0 \leftarrow X_0^i$
6: $\quad \hat{Y}_0^i \leftarrow [\hat{X}_0^T, H]^T$
7: $\quad$ **for** $t$ in $\{0, 1, \cdots, H-1\}$ **do**
8: $\quad\quad$ **if** $C(\hat{X}_t)$ occurs **then**
9: $\quad\quad\quad \hat{X}_{t+1} \leftarrow X_{t+1}^i$
10: $\quad\quad$ **else**
11: $\quad\quad\quad \hat{X}_{t+1} \leftarrow \hat{X}_t$
12: $\quad\quad$ **end if**
13: $\quad\quad \hat{Y}_{t+1}^i \leftarrow [\hat{X}_t^T, H-t-1]^T$
14: $\quad$ **end for**
15: $\quad \tilde{\mathcal{D}}_i \leftarrow \{\hat{Y}_{0:H}^i, U_{0:H}^i\}$
16: $\quad \tilde{\mathcal{D}} \leftarrow \tilde{\mathcal{D}} \cup \{\tilde{\mathcal{D}}_i\}$
17: **end for**
18: **Return** $\tilde{\mathcal{D}}$

---

We define an observable mediator variable $M_t \in \mathcal{M}$[8], consider the spaces $\mathcal{X}, \mathcal{U}, \mathcal{W}$ and $\mathcal{M}$ to be discrete, and make the following assumption.

**Assumption 3.5.** The mediator $M_t$ intercepts every directed path from $U_t$ to $S_{t+1}$. The observable state $X_t$ blocks all back-door paths from $U_t$ to $M_t$. All back-door paths from $M_t$ to $X_{t+1}$ are blocked by $(X_t, U_t)$.

Here, the definitions for directed path and back-door path follow the definitions in Pearl 2009, Chapter 3.3.2. We also

---

[8]Since $M_t$ is observable, in this example, the sequence $M_{0:H}^i, i \in \{1, 2, \cdots, N_D\}$ is also in the offline dataset $\mathcal{D}$.

---

define the Q-function conditioned on the mediator as

$$Q_M^\pi([x^T, k]^T, u, m) :=$$

$$\mathbb{E}_{\tilde{\mathcal{P}}_{\text{online}}}[\sum_{\tau=0}^{k} r(\hat{Y}_\tau)|\hat{Y}_0 = [x^T, k]^T, U_0 = u, M_0 = m, \pi]$$

(40)

$$\mathbb{E}_{\tilde{\mathcal{P}}_{\text{online}}}[\sum_{\tau=0}^{\infty} r(\hat{Y}_\tau)|\hat{Y}_0 = [x^T, k]^T, U_0 = u, M_0 = m, \pi].$$

(41)

The Bellman equation is given by

$$Q_M^\pi(\hat{Y}_t, U_t, M_t)$$
$$= r(\hat{Y}_t) + \mathbb{E}_{\tilde{\mathcal{P}}_{\text{online}}(\hat{Y}_{t+1}|\hat{Y}_t, U_t, M_t)}[V^\pi(\hat{Y}_{t+1})|\hat{Y}_t, U_t, M_t]$$

(42)

$$= r(\hat{Y}_t) + \mathbb{E}_{\tilde{\mathcal{P}}_{\text{offline}}(\hat{Y}_{t+1}|\hat{Y}_t, U_t, M_t)}[V^\pi(\hat{Y}_{t+1})|\hat{Y}_t, U_t, M_t],$$

(43)

where (43) holds because $\tilde{\mathcal{P}}_{\text{online}}(\hat{Y}_{t+1}|\hat{Y}_t, U_t, M_t) = \tilde{\mathcal{P}}_{\text{offline}}(\hat{Y}_{t+1}|\hat{Y}_t, U_t, M_t)$ ($\tilde{\mathcal{P}}_{\text{online}}(\hat{Y}_{t+1}|\hat{Y}_t, U_t, M_t)$ can be consistently estimated from offline data (Pearl, 2009)). Due to (43), we can estimate $Q_M^\pi$ iteratively with data $\tilde{\mathcal{D}}$ by solving

$$\arg\min_{Q \in \mathcal{Q}} \sum_{i=1}^{N_D} \sum_{t=0}^{H-1} \left(r(\hat{Y}_t^i) - Q(\hat{Y}_t^i, U_t^i, M_t^i) + \hat{V}^l(\hat{Y}_{t+1}^i)\right)^2$$

(44)

at iteration $l+1$, where $\mathcal{Q}$ is the class of functions of the form $\mathcal{Y} \times \mathcal{U} \times \mathcal{M} \rightarrow [0, 1]$, and $\hat{V}^l$ is the estimation for the value function $V^\pi$ in iteration $l$. To evaluate (44), one needs to evaluate $V^\pi$ using $Q_M^\pi$ and known offline statistics. The value function can be written as

$$V^\pi(y)$$
$$= \sum_{u \in \mathcal{U}} Q^\pi(y, u)\pi(U_t = u|\hat{Y}_t = y)$$

(45)

$$= \sum_{u \in \mathcal{U}} \mathbb{E}_{\tilde{\mathcal{P}}_{\text{online}}(\hat{Y}_{t+1}|\hat{Y}_t, U_t)}[V^\pi(\hat{Y}_{t+1})$$
$$+ r(y)|\hat{Y}_t = y, U_t = u]\pi(U_t = u|\hat{Y}_t = y)$$

(46)

$$= \sum_{u \in \mathcal{U}} \sum_{y' \in \mathcal{Y}} (V^\pi(y') + r(y))$$
$$\tilde{P}_{\text{online}}(\hat{Y}_{t+1} = y'|\hat{Y}_t = y, U_t = u)\pi(U_t = u|\hat{Y}_t = y),$$

(47)

where (46) is due to the Bellman equation. From the front-door adjustment formula in Pearl 2009, Chapter 3.3.2, con-

ditioned on $\hat{Y}_t$, we have

$$\tilde{P}_{\text{online}}(\hat{Y}_{t+1} = y'|U_t = u, \hat{Y}_t = y) =$$
$$\sum_{m \in \mathcal{M}} \sum_{u' \in \mathcal{U}} \tilde{P}_{\text{offline}}(\hat{Y}_{t+1} = y'|U_t = u', M_t = m, \hat{Y}_t = y)$$
$$\tilde{P}_{\text{offline}}(U_t = u'|\hat{Y}_t = y)\tilde{P}_{\text{offline}}(M_t = m|U_t = u, \hat{Y}_t = y) \tag{48}$$

given Assumption 3.5. Substituting into (47), we have

$$V^\pi(y)$$
$$= \sum_{u \in \mathcal{U}} \sum_{y' \in \mathcal{Y}} (V^\pi(y') + r(y))$$
$$\sum_{m \in \mathcal{M}} \sum_{u' \in \mathcal{U}} \tilde{P}_{\text{offline}}(\hat{Y}_{t+1} = y'|U_t = u', M_t = m, \hat{Y}_t = y)$$
$$\tilde{P}_{\text{offline}}(U_t = u'|\hat{Y}_t = y)\tilde{P}_{\text{offline}}(M_t = m|U_t = u, \hat{Y}_t = y)$$
$$\pi(U_t = u|\hat{Y}_t = y) \tag{49}$$
$$= \sum_{m \in \mathcal{M}} \sum_{u' \in \mathcal{U}} \sum_{u \in \mathcal{U}} \sum_{y' \in \mathcal{Y}} (V^\pi(y') + r(y))$$
$$\tilde{P}_{\text{offline}}(\hat{Y}_{t+1} = y'|U_t = u', M_t = m, \hat{Y}_t = y)$$
$$\tilde{P}_{\text{offline}}(U_t = u'|\hat{Y}_t = y)\tilde{P}_{\text{offline}}(M_t = m|U_t = u, \hat{Y}_t = y)$$
$$\pi(U_t = u|\hat{Y}_t = y). \tag{50}$$

Similar to (47), from Bellman equation (43), we can write $Q_M^\pi$ as

$$Q_M^\pi(y, u, m) = \sum_{y' \in \mathcal{Y}} (V^\pi(y') + r(y))$$
$$\tilde{P}_{\text{offline}}(\hat{Y}_{t+1} = y'|U_t = u, M_t = m, \hat{Y}_t = y). \tag{51}$$

Substituting into (50), we have

$$V^\pi(y)$$
$$= \sum_{m \in \mathcal{M}} \sum_{u' \in \mathcal{U}} \sum_{u \in \mathcal{U}} Q_M^\pi(y, u', m)\tilde{P}_{\text{offline}}(U_t = u'|\hat{Y}_t = y)$$
$$\tilde{P}_{\text{offline}}(M_t = m|U_t = u, \hat{Y}_t = y)\pi(U_t = u|\hat{Y}_t = y), \tag{52}$$

whose RHS only includes $Q_M^\pi$ and distributions in the offline statistics. It is also easy to obtain $Q^\pi$ using $Q_M^\pi$. We have

$$Q^\pi(y, u)$$
$$= \mathbb{E}_{\tilde{\mathcal{P}}_{\text{online}}(M_t|U_t, \hat{Y}_t)}[Q_M^\pi(\hat{Y}_t, U_t, M_t)|\hat{Y}_t = y, U_t = u] \tag{53}$$
$$= \mathbb{E}_{\tilde{\mathcal{P}}_{\text{offline}}(M_t|U_t, \hat{Y}_t)}[Q_M^\pi(\hat{Y}_t, U_t, M_t)|\hat{Y}_t = y, U_t = u] \tag{54}$$

because $\tilde{\mathcal{P}}_{\text{online}}(M_t|U_t, \hat{Y}_t) = \tilde{\mathcal{P}}_{\text{offline}}(M_t|U_t, \hat{Y}_t)$. We introduce the proposed algorithm in Algorithm 2. To ensure the safety objective while preserving as much performance

as possible, we use the following optimization problem to obtain the safe control action:

$$\arg\min_{u \in \mathcal{U}} J(U^n, u) \tag{55}$$
$$\text{s.t. } S(X_t, u, t) \geq 0,$$

where $U^n$ is the control action obtained from the policy $\pi^n$ and $J : \mathcal{U} \times \mathcal{U} \to \mathbb{R}$ is a function that penalizes deviation from $U^n$.

---

**Algorithm 2** Proposed control algorithm

1: **Require:** offline dataset $\mathcal{D}$
2: Obtain dataset $\tilde{\mathcal{D}}$ with dataset $\mathcal{D}$ and Algorithm 1
3: $l \leftarrow 0$
4: Initialize $\hat{Q}_M^{\pi,0} \in \mathcal{Q}$
5: **while** not converged **do**
6:     $\hat{Q}_M^{\pi,l+1} \leftarrow$ (44)
7:     $l \leftarrow l + 1$
8: **end while**
9: $Q_M^\pi \leftarrow \hat{Q}_M^{\pi,l}$
10: $t \leftarrow 0$
11: **while** $t < H$ **do**
12:     observe state $X_t$
13:     $U^n \sim \pi^n(\cdot|X_t)$
14:     Estimate $Q^\pi$ using (54) with $Q_M^\pi$
15:     $U_t \leftarrow$ (55)
16:     execute control action $U_t$
17:     $t \leftarrow t + 1$
18: **end while**

---

## 4. Numeric Simulation

We consider a setting that resembles a simplified driving scenario with discrete state space. Let $X_t = [X_t^1, X_t^2]^T \in \mathbb{Z}^2$ be the state of the system, where $X_t^1$ represents the position of the vehicle on a 1-dimensional road, and $X_t^2$ represents the velocity of the vehicle. The control action $U_t \in \{-3, -2, -1, 0, 1\}$ represents the acceleration or deceleration applied to the wheels. The latent variable $W_t \in \{0, 1, 2, 3\}$ represents the slipperiness of the road, which can make the acceleration or deceleration applied to the wheels less effective. The system also has uncertainty $N_t = [N_t^1, N_t^2]^T \in \{-1, 0, 1\} \times \{-2, -1, 0, 1, 2\}$. The system transition is given by

$$X_{t+1}^1 = X_t^1 + X_t^2 \tag{56}$$
$$X_{t+1}^2 = \max(0, X_t^2 +$$
$$\text{sign}(U_t + N_t^1)\max(0, |U_t + N_t^1| - W_t) + N_t^2). \tag{57}$$

The distributions of $W_t$ and $N_t$ are given in Appendix C. The safety requirement is that the vehicle obeys a varying

speed limit. Specifically,

$$C(X_t) = \{\mathbf{1}\{X_t^1 \mod 10 < 4\} \cap \{X_t^2 \le 3\}\}$$
$$\cup \{\mathbf{1}\{X_t^1 \mod 10 \ge 4\} \cap \{X_t^2 \le 5\}\}. \quad (58)$$

The offline dataset can be considered as human driving dataset where the human observes the slipperiness of the road in their behavioral policy, but the slipperiness is not recorded by the sensor. Specifically, we consider a behavioral policy $\pi^b$ that applies heavier brakes when the road is more slippery. The detailed distribution for the behavioral policy is given in Appendix C. The nominal policy simply chooses actions in the action space with identical probability, *i.e.*, $\pi(U_t|X_t) = 0.2, \forall U_t \in \{-3, -2, -1, 0, 1\}, X_t \in \mathbb{Z}^2$. We consider $H = 10$ and $\epsilon = 0.2$. We run 100 simulations, where each simulation simulates 100 trajectories starting from $X_0 = [0, 0]^T$, with the following 2 methods:

1. The proposed method. The proposed method has access to an unbiased estimate for the Q-function $Q^\pi$, which can be estimated using causal reinforcement learning method such as Wang et al. 2021b and Shi et al. 2024.

2. The discrete-time control barrier function (DTCBF) proposed in Cosner et al. 2023. This method cannot utilize the Q-function, so the safety condition is evaluated using the distribution obtained from the offline dataset. The detailed parameters are given in Appendix C.

For both methods, the control policy is to maximize the control action while ensuring the corresponding safety condition.

The simulation results are illustrated in Figure 1 and 2. The results show that the proposed controller, although having no access to the latent variable $W$ or the ground truth transition dynamics, can achieve a safety performance that satisfies the safety objective (6) with the Q-function. On the other hand, the discrete-time control barrier function cannot achieve the safety objective with the offline statistics even if the control action satisfies the safety condition.

## 5. Conclusion and future work

While there may be a common belief that distribution shifts only happen when system dynamics change, spurious distribution shifts can arise due to the use of observed statistics with latent variables. In this paper, we introduce a novel approach for learning safety certificates using only observed statistics with such distribution shifts.

To our knowledge, the proposed work is also the first to apply causal reinforcement learning in the design of safety certificates. The framework leverages a new notion of invariance—probabilistic invariance—that enables a safety

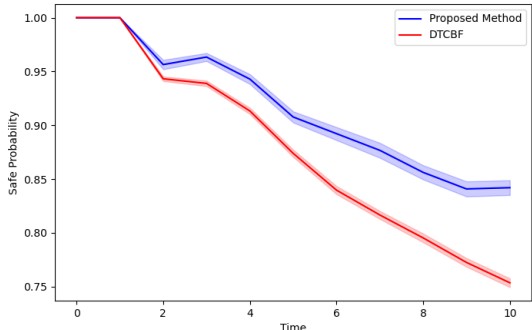

Figure 1. Probability of safety at each time for both controllers with 95% confidence interval shown in the shady region.

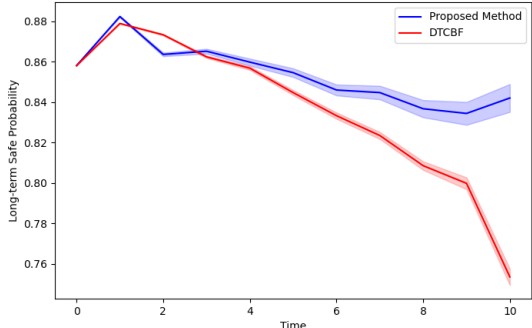

Figure 2. Long-term safety at each time for both controllers with 95% confidence interval shown in the shady region. The long-term safety is equal to $\mathbb{P}^{\hat{\pi}, \pi}(C(X_t) \cap C(X_{t+1}) \cap \cdots \cap C(X_H)|X_0)$ defined in (6).

certificate to guarantee long-term safety with persistent feasibility, even when the system's dynamics are only partially identifiable. Beyond the proposed algorithm, we expect the proposed framework can be extended to incorporate other causal inference techniques for the design of safety certificates.

Furthermore, while many techniques are developed for probabilistic reachability, its standard definitions involving all states may not be computable from observed statistics without complete system models (see Section 2.2). On the other hand, in Theorem 3.2 and Theorem 3.4, such definitions of reachability may not be necessary for the purpose of safe control: For example, the value-like function, learnable from observed statistics, can be used to guarantee safety and persistent feasibility. This suggests a future direction: to explore new measures of reachability and latent risk that are both learnable from observed data and sufficient for safety certification.

## Acknowledgment

This work is sponsored in part by Carnegie Mellon University Security and Privacy Institute (CyLab), in part by the PRESTO Grant Number JPMJPR2136 from Japan Science and Technology Agency, in part by National Science Foundation under Grant No. 2442948, and in part by the Department of the Navy, Office of Naval Research, under award number N00014-23-1-2252. The views expressed are those of the authors and do not reflect the official policy or position of the US Navy, Department of Defense or the US Government.

## Impact Statement

This paper presents work whose goal is to advance the field of Machine Learning. There are many potential societal consequences of our work, none which we feel must be specifically highlighted here.

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

## A. Mismatch between Online Statistics and Offline Statistics: An Example

Consider a system with observable state $X_t \in \{0,1\}$ and latent variable $W_t \in \{0,1\}$. The control action is $U_t \in \{0,1\}$. The system is considered to be safe when $X_t = 0$. Suppose that the state transition probabilities are given by

$$\mathbb{P}(X_{t+1} = 0|X_t = 0, W_t = 0, U_t = 0) = 0.9 \tag{59}$$

$$\mathbb{P}(X_{t+1} = 0|X_t = 0, W_t = 1, U_t = 0) = 1 \tag{60}$$

$$\mathbb{P}(X_{t+1} = 0|X_t = 1, W_t = 1, U_t = 0) = 0 \tag{61}$$

$$\mathbb{P}(X_{t+1} = 0|X_t = 0, W_t = 0, U_t = 1) = 1 \tag{62}$$

$$\mathbb{P}(X_{t+1} = 0|X_t = 0, W_t = 1, U_t = 1) = 0.1 \tag{63}$$

$$\mathbb{P}(X_{t+1} = 0|X_t = 1, W_t = 1, U_t = 1) = 0. \tag{64}$$

The value for $W_t$ is determined by

$$\mathbb{P}(W_t = 0|X_t = 0) = 0.5 \tag{65}$$

$$\mathbb{P}(W_t = 0|X_t = 1) = 0. \tag{66}$$

Note that $\mathbb{P}(X_{t+1} = 1|\cdot) = 1 - \mathbb{P}(X_{t+1} = 0|\cdot)$, and we omit the case for $X_t = 1, W_t = 0$ because it is not reachable from other states. The behavioral policy $\pi^b$ is given by

$$\pi^b(U_t = 0|X_t = 0, W_t = 0) = 0.5 \tag{67}$$

$$\pi^b(U_t = 0|X_t = 0, W_t = 1) = 1. \tag{68}$$

We find out the safe probability of the immediate next time step given $U_t = 1$ for both the offline statistics and the online statistics. We have

$$\mathbb{P}_{\text{offline}}(X_{t+1} = 0|X_t = 0, U_t = 1) = \frac{\mathbb{E}_{W_t \sim \mathbb{P}(W_t|X_t)}[\mathbb{P}(X_{t+1} = 0|X_t = 0, W_t, U_t = 1)\pi^b(U_t = 1|X_t = 0, W_t)]}{\mathbb{E}_{W_t \sim \mathbb{P}(W_t|X_t)}[\mathbb{P}(U_t = 1|X_t = 0, W_t)]} \tag{69}$$

$$= 1 \tag{70}$$

and

$$\mathbb{P}_{\text{online}}(X_{t+1} = 0|X_t = 0, U_t = 1) = \mathbb{E}_{W_t \sim \mathbb{P}(W_t|X_t)}[\mathbb{P}(X_{t+1} = 0|X_t = 0, W_t, U_t = 1)] \tag{71}$$

$$= 0.55. \tag{72}$$

We observe that the safe probability evaluated from the online statistics is significantly lower than the safe probability evaluated from the offline statistics. This shows that if a controller uses the offline statistics to perform safe control, the safe probabilities associated with some control actions will be significantly over-approximated.

## B. Proof of Proposition 3.1

*Proof.* Consider a function $\tilde{\Psi}^\pi : \mathcal{X} \times \mathbb{Z} \to [0,1]$ defined as

$$\tilde{\Psi}^\pi(x,t) := \tilde{\mathbb{P}}^\pi(C(\hat{X}_t) \cap C(\hat{X}_{t+1}) \cap \cdots \cap C(\hat{X}_H)|\hat{X}_t = x), \tag{73}$$

where the probability is evaluated with the assumption that the sequence $\hat{X}_{t:H}$ has the statistics (8). We first show that

$$\tilde{\Psi}^\pi(x,t) = \Psi^\pi(x,t), \forall x \in \mathcal{X}, t \in \{0, 1, \cdots, H\}. \tag{74}$$

We have

$$\Psi^\pi(x,t) = \int_{\mathcal{X}^{H-t+1}} \mathbf{1}\{C(X_t) \cap C(X_{t+1}) \cap \cdots \cap C(X_H)\} P^\pi(X_{t:H}|X_t = x)dX_{t:H}, \tag{75}$$

where $P^\pi(X_{t:H}|X_t = x)$ is the conditional distribution of the sequence $X_{t:H}$ given $X_t = x$ when the sequence has statistics (4) and a policy $\pi$ is used. Similarly, we have

$$\tilde{\Psi}^\pi(x,t) = \int_{\mathcal{X}^{H-t+1}} \mathbf{1}\{C(\hat{X}_t) \cap C(\hat{X}_{t+1}) \cap \cdots \cap C(\hat{X}_H)\} \tilde{P}_x^\pi(\hat{X}_{t:H}|\hat{X}_t = x)d\hat{X}_{t:H}, \tag{76}$$

where $\tilde{P}_x^\pi(\hat{X}_{t:H}|\hat{X}_t = x)$ is the conditional distribution of the sequence $\hat{X}_{t:H}$ given $\hat{X}_t = x$ when the sequence has statistics (8) and a policy $\pi$ is used. Note that, when the sequences $X_{t:H}$ and $\hat{X}_{t:H}$ take the same value, $P^\pi(X_{t:H}|X_t = x) \neq \tilde{P}_x^\pi(\hat{X}_{t:H}|\hat{X}_t = x)$ only if there exists a time $\tau \in \{t, t+1, \cdots, H\}$ such that $C(X_\tau)$ (and $C(\hat{X}_\tau)$ since $X_\tau = \hat{X}_\tau$) does not occur. In such case, we have $\mathbf{1}\{C(X_t) \cap C(X_{t+1}) \cap \cdots \cap C(X_H)\} = \mathbf{1}\{C(\hat{X}_t) \cap C(\hat{X}_{t+1}) \cap \cdots \cap C(\hat{X}_H)\} = 0$. Therefore, we have (74). Next, we show that

$$\tilde{\Psi}^\pi(x, H-k) = V^\pi([x^T, k]^T), \forall x \in \mathcal{X}, k \in \{0, 1, \cdots, H\}. \tag{77}$$

We have

$$V^\pi([x^T, k]^T) = \int_{\mathcal{Y}^{k+1}} \left(\sum_{\tau=0}^k r([\hat{X}_\tau^T, K_\tau]^T)\right) \tilde{P}^\pi(\hat{Y}_{0:k}|\hat{Y}_0 = [x^T, k]^T) d\hat{Y}_{0:k}, \tag{78}$$

where $\tilde{P}^\pi(\hat{Y}_{0:k}|\hat{Y}_0 = [x^T, k]^T)$ is the conditional distribution of the sequence $\hat{Y}_{0:k}$ given $\hat{Y}_0 = [x^T, k]^T$ when the sequence has statistics $\tilde{\mathcal{P}}_{\text{online}}(\hat{Y}_{t+1}|\hat{Y}_t, U_t)$ and a policy $\pi$ is used. Since $r([x^T, k]^T) \neq 0$ only if $k = 0$, and $K_\tau = 0$ when $\tau = k$ given $K_0 = k$, we have

$$V^\pi([x^T, k]^T) = \int_{\mathcal{Y}^{k+1}} r([\hat{X}_k^T, 0]^T) \tilde{P}^\pi(\hat{Y}_{0:k}|\hat{Y}_0 = [x^T, k]^T) d\hat{Y}_{0:k}. \tag{79}$$

Since the distribution of sequence $X_{0:k}$ and the distribution of sequence $K_{0:k}$ are independent, we have

$$V^\pi([x^T, k]^T) = \int_{\mathcal{X}^{k+1}} r([\hat{X}_k^T, 0]^T) \tilde{P}_x^\pi(\hat{X}_{0:k}|\hat{X}_0 = x) d\hat{X}_{0:k}. \tag{80}$$

From (76), we have

$$\tilde{\Psi}^\pi(x, H-k) = \int_{\mathcal{X}^{k+1}} \mathbf{1}\{C(\hat{X}_{H-k}) \cap C(\hat{X}_{H-k+1}) \cap \cdots \cap C(\hat{X}_H)\} \tilde{P}_x^\pi(\hat{X}_{H-k:H}|\hat{X}_{H-k} = x) d\hat{X}_{H-k:H} \tag{81}$$

$$= \int_{\mathcal{X}^{k+1}} \mathbf{1}\{C(\hat{X}_0) \cap C(\hat{X}_1) \cap \cdots \cap C(\hat{X}_k)\} \tilde{P}_x^\pi(\hat{X}_{0:k}|\hat{X}_0 = x) d\hat{X}_{0:k}. \tag{82}$$

Note that $r([\hat{X}_k^T, 0]^T) = 1$ iff $C(\hat{X}_\tau)$ occurs for all $\tau \in \{0, 1, \cdots, k\}$, which gives $r([\hat{X}_k^T, 0]^T) = \mathbf{1}\{C(\hat{X}_0) \cap C(\hat{X}_1) \cap \cdots \cap C(\hat{X}_k)\}$. Therefore, we have (77). $\square$

## C. Details in Simulation

The distribution of $W_t$ is given by

$$\mathbb{P}(W_t = 0) = \mathbb{P}(W_t = 1) = \frac{1}{2} \tag{83}$$

if $X_t^1 \mod 6 \geq 3$ and

$$\mathbb{P}(W_t = 1) = \mathbb{P}(W_t = 2) = \mathbb{P}(W_t = 3) = \frac{1}{3} \tag{84}$$

if $X_t^1 \mod 6 < 3$. The distribution of $N_t$ is given by

$$\mathbb{P}(N_t^1 = -1) = \mathbb{P}(N_t^1 = 0) = \mathbb{P}(N_t^1 = 1) = \frac{1}{3} \tag{85}$$

and

$$\mathbb{P}(N_t^2 = -2) = \mathbb{P}(N_t^2 = -1) = \mathbb{P}(N_t^2 = 0) = \mathbb{P}(N_t^2 = 1) = \mathbb{P}(N_t^2 = 2) = \frac{1}{5}. \tag{86}$$

The behavioral policy $\pi^b$ is defined as follows. When $W_t \geq 1$, $X_t^1 \mod 10 < 4$, and $X_t^2 \geq 2$, the policy satisfies

$$\pi^b(U_t = -3 | X_t, W_t) = 0.5 \tag{87}$$
$$\pi^b(U_t = -2 | X_t, W_t) = 0.4 \tag{88}$$
$$\pi^b(U_t = -1 | X_t, W_t) = 0.05 \tag{89}$$
$$\pi^b(U_t = 0 | X_t, W_t) = 0.04 \tag{90}$$
$$\pi^b(U_t = 1 | X_t, W_t) = 0.01. \tag{91}$$

When $W_t \geq 1$, $X_t^1 \mod 10 \geq 4$, and $X_t^2 \geq 4$, the policy satisfies

$$\pi^b(U_t = -3 | X_t, W_t) = 0.5 \tag{92}$$
$$\pi^b(U_t = -2 | X_t, W_t) = 0.4 \tag{93}$$
$$\pi^b(U_t = -1 | X_t, W_t) = 0.05 \tag{94}$$
$$\pi^b(U_t = 0 | X_t, W_t) = 0.04 \tag{95}$$
$$\pi^b(U_t = 1 | X_t, W_t) = 0.01. \tag{96}$$

When $W_t \geq 2$, $X_t^1 \mod 10 < 4$, and $X_t^2 \geq 1$, the policy satisfies

$$\pi^b(U_t = -3 | X_t, W_t) = 0.5 \tag{97}$$
$$\pi^b(U_t = -2 | X_t, W_t) = 0.4 \tag{98}$$
$$\pi^b(U_t = -1 | X_t, W_t) = 0.05 \tag{99}$$
$$\pi^b(U_t = 0 | X_t, W_t) = 0.04 \tag{100}$$
$$\pi^b(U_t = 1 | X_t, W_t) = 0.01. \tag{101}$$

When $W_t \geq 2$, $X_t^1 \mod 10 \geq 4$, and $X_t^2 \geq 3$, the policy satisfies

$$\pi^b(U_t = -3 | X_t, W_t) = 0.5 \tag{102}$$
$$\pi^b(U_t = -2 | X_t, W_t) = 0.4 \tag{103}$$
$$\pi^b(U_t = -1 | X_t, W_t) = 0.05 \tag{104}$$
$$\pi^b(U_t = 0 | X_t, W_t) = 0.04 \tag{105}$$
$$\pi^b(U_t = 1 | X_t, W_t) = 0.01. \tag{106}$$

When $W_t \geq 3$, $X_t^1 \mod 10 < 4$, and $X_t^2 \geq 2$, the policy satisfies

$$\pi^b(U_t = -3 | X_t, W_t) = 0.9 \tag{107}$$
$$\pi^b(U_t = -2 | X_t, W_t) = 0.05 \tag{108}$$
$$\pi^b(U_t = -1 | X_t, W_t) = 0.03 \tag{109}$$
$$\pi^b(U_t = 0 | X_t, W_t) = 0.01 \tag{110}$$
$$\pi^b(U_t = 1 | X_t, W_t) = 0.01. \tag{111}$$

When $W_t \geq 3$, $X_t^1 \mod 10 \geq 4$, and $X_t^2 \geq 4$, the policy satisfies

$$\pi^b(U_t = -3 | X_t, W_t) = 0.9 \tag{112}$$
$$\pi^b(U_t = -2 | X_t, W_t) = 0.05 \tag{113}$$
$$\pi^b(U_t = -1 | X_t, W_t) = 0.03 \tag{114}$$
$$\pi^b(U_t = 0 | X_t, W_t) = 0.01 \tag{115}$$
$$\pi^b(U_t = 1 | X_t, W_t) = 0.01. \tag{116}$$

Otherwise, the policy satisfies

$$\pi^b(U_t = -3|X_t, W_t) = \pi^b(U_t = -2|X_t, W_t) = \pi^b(U_t = -1|X_t, W_t) = \pi^b(U_t = 0|X_t, W_t)$$
$$= \pi^b(U_t = 1|X_t, W_t) = 0.2. \tag{117}$$

For the discrete-time control barrier function, we first represent the safety requirement using $C(X_t) = \mathbf{1}\{X_t \in \mathcal{C}\}$, where $\mathcal{C} = \{x \in \mathbb{Z}^2 : h(x) \geq 0\}$, and

$$h([x^1, x^2]^T) = \tanh\left(4.5 + \sum_{n \in \{1,3,5,7\}} \frac{4}{n\pi} \sin(-\frac{\pi}{5}n(x^1 + 0.5)) - x^2\right). \tag{118}$$

We use the safety condition

$$\mathbb{E}[h(X_{t+1})|X_t, U_t] \geq \alpha h(X_t) + \delta \tag{119}$$

with $\alpha = 0.01$ and $\delta = -0.5$, such that the condition (6) is guaranteed for $\epsilon = 0.2$ due to Cosner et al. 2023, equation (13).

