# OpenReview forum: "Safety Certificate against Latent Variables with Partially Unidentifiable Dynamics"
_ICML.cc/2025/Conference — ICML 2025 poster_

### Official Review · Reviewer_aHyV · 2025-03-10

**Overall Recommendation:** 3

**Summary:**

This paper proposes a safety certificate against latent/unobservable latent variables that cause a distribution shift between offline learning data and online execution. The paper aims to efficiently ensure long-term safety for stochastic systems. The safety certificate in probabilistic space can construct a forward invariable condition that can be implemented in real-time control. The authors validate the proposed approach on a naive 2-dimensional numerical example, again the control barrier function approach.

**Claims And Evidence:**

Yes, the reviewer acknowledges that this paper and its authors make clear and convincing claims.

**Essential References Not Discussed:**

The safety certificate formulated in this paper (key contribution) looks very similar to the one in
Enforcing Hard Constraints with Soft Barriers: Safe Reinforcement Learning in Unknown Stochastic Environments, ICML 2023, and
Stochastic Safety Verification Using Barrier Certificates, CDC 2004
where they used the super-martingale property to formulate a barrier certificate for a probabilistic safety guarantee.

The authors may have to discuss how their approach compares to the existing references.

**Experimental Designs Or Analyses:**

The experiment is sound. However, the major concern is the experiment design is only for a 2-dimensional numerical system, which is not convincing.

**Methods And Evaluation Criteria:**

The evaluation criteria make sense. It includes the unobservable latent variable that causes distribution shifts, and the evaluations are with the long-term safety probability.

**Other Comments Or Suggestions:**

N/A

**Other Strengths And Weaknesses:**

As mentioned above, strengths include studying an important problem for the community (well-motivated), most of the techniques and theoretical contributions are sound (from the part I can understand).

Weaknesses include writing clarity, some major missing discussion on the existing references, and weak experiments.

**Questions For Authors:**

1. How strong is assumption 2.1? and what does this assumption mean? Is this assumption common in real-world applications?
2. In eq3, why the behavioral policy can depend on W_t, while the online policy cannot? The reviewer cannot figure out the fundamental difference between them.
3. Please elaborate more on Eq8, what's the intuition of its design and what's the purpose?
4. Eq15 looks like the super-martingale property used in barrier certificates in stochastic systems, as listed in the above two references.
5. Wht r(\hat{Y}_t) = 0 (line 286)?
6. Can you make the section after assumption 3.5 clearer (how the mediator works and how the Q function is obtained/learned)?

**Relation To Broader Scientific Literature:**

This paper is related to the safe control, safe reinforcement learning community. The safety certificate with the unobservable latent dynamics is interesting and important to the community.

**Theoretical Claims:**

Yes, but not all the theoretical claims. The reviewer has checked Section 3.2, safety condition, Section 3.3 Evaluation of Safety Condition and Persistent Feasibility Guarantee, and didn't find any issue.

---

> ### Author Rebuttal · Authors · 2025-04-01
>
> We sincerely appreciate your efforts and valuable feedback. Please find below our response to your comments.
>
> References Not Discussed:
>
> Wang et al. 2023
>
> We will cite this paper. This paper studies the problem of training a safe policy, to be continuously used, when the data with complete state is used for training, and the control policy also has access to the complete state. Our problem differs from this paper as follows.
> First, since the latent variable cannot be observed, the training cannot use data with complete state, and the policy cannot take the full state as input. Estimation of system dynamics or value functions using observable data only will be biased due to the distribution shifts between online vs. offline statistics arising from latent variables, even if the available data is infinite.
> Second, our problem characterizes the set of safe control actions, which is useful for scenarios such as when the system already has a nominal controller. Since any action in the set of safe control actions is allowed, our problem requires additional steps to derive persistent feasibility (section 3.3), in order to achieve long-term safety. On the other hand, when an optimal policy is continuously used, the feasibility of actions can be directly obtained from the optimization problem used to compute the optimal policy.
>
> Prajna et al. 2004, and response to Q4
>
> We have cited the journal version of this paper (Prajna et al. 2007). Both barrier certificates and proposed methods use conditions, reminiscent of super-martingale, to control the evaluation of certain values that do not decrease over time. Barrier certificates can directly formulate this condition on the Barrier function since the complete system dynamics is known and all states are observed. On the other hand, we need to define two auxiliary stochastic processes and value-like functions for the observed statistics, find the relation between safety probability and value-like functions in order to use causal RL methods to adjust the impact of latent variables by borrowing tools from causal techniques. This is the first paper that derives how to convert the problem of assuring long-term safety probability, with auxiliary stochastic processes and new definition of value-like functions, into the form that can be handled by causal RL techniques.
>
> Q1, Q2.
> From Assumption 2.1, the distribution of latent variable $W_t$ can depend on the current observed variable $X_t$, but conditioned on this dependence, it doesn’t depend on the history of the variables.
> Take an example when an object may jump from occluded areas: $W_t$ can denote the presence of an occluded object. The likelihood depends on the ego-vehicle’s location (included in the observed state), but given this location, it doesn’t depend on the past trajectories of the ego-vehicle or other occluded objects along the past.
> This formulation of behavioral policy allows the offline data to be based on a behavioral policy that is different from the online policy. The online policy can only use the observed state, while the behavioral policy is also allowed to depend on different sets of information/measurement than the online policy.
> The models are studied as confounded MDP, and more examples can be found in papers such as Shi et al. 2024 and Wang et al. 2021.
>
> Q3.
> The design cannot directly use the underlying decision process because we cannot evaluate the standard value/Q-functions which are functions of all states (latent variables and observed variables), but naively defining a value-like function using observed variables results in a wrong safety estimate due to the distribution shifts between offline vs. online statistics. Instead, we construct two auxiliary stochastic processes in Eq. 8 (for online statistics) and Eq. 11 (for offline statistics) from the underlying decision process, allowing us to explicitly handle the distribution shifts between offline vs. online statistics.
>
> Q5.
> In order for the safety probability to match the Q-function, the reward needs to be defined as Eq. 10. Eq. 10 leads to $r(\hat{Y}_t) = 0$
>
> Q6.
> We apologize for the confusion. The method that learns Q-function using the mediator is proposed in the existing work (Shi et al. 2024), which is only an example to demonstrate how the proposed method can be integrated with existing methods that learn Q-function, and is not considered as our contribution. In addition, we would like to clarify that Assumption 3.5, together with the introduction of the mediator variable, is subject to this example only and not to the proposed method, and the proposed method works with any method that learns unbiased Q-function. Regarding the details of how the mediator works and how the Q-function is learned, due to space limitations here and in the paper, please refer to Section 3.6 of Shi et al. 2024 for more detailed explanations and derivations.

---

### Official Review · Reviewer_toZQ · 2025-03-14

**Overall Recommendation:** 3

**Summary:**

The paper addresses the challenge of ensuring long-term safety in control systems that have latent (unobserved) variables causing partially unidentifiable dynamics and distribution shifts between offline (training) and online (deployment) data​. Traditional safety assurance methods typically assume full knowledge of the system or perfect state observability, which is often unrealistic when latent factors are present​. In response, this work proposes a probabilistic safety certificate framework that operates under partial observability. The key idea is to formulate forward invariance conditions in probability space, rather than in state-space, so that safety can be quantified using observed state statistics even under distribution shifts induced by latent variables.

The methodology integrates concepts from stochastic safe control and causal reinforcement learning. The authors derive conditions (safety constraints) on control actions that guarantee a minimum probability of remaining in a safe set for the entire horizon (episode) despite latent uncertainties​. They establish a relationship between a risk measure (long-term safe probability) and a suitably defined marginalized value/Q-function via a modified Bellman equation​. By leveraging this relationship, the safety certificate enforces that at every time step there exists at least one feasible action that keeps the system’s risk below a specified tolerance (e.g. ensuring safety probability ≥ 1–ε)​. A real-time controller can then use these action constraints: essentially acting myopically (greedily) but only choosing actions that satisfy the long-term safety certificate. The approach is designed to be computationally efficient for online use​. Importantly, the framework uses causal RL techniques to estimate the necessary Q-functions from offline data despite latent confounders, marking (as claimed) the first integration of causal RL for quantifying long-term risk in safety certificate design​. The paper demonstrates the effectiveness of the proposed safety certificate through numerical simulations of a driving scenario with an unobserved “road slipperiness” factor, showing that the new method maintains safety where prior methods fail​

**Claims And Evidence:**

The authors make several clear claims: (i) that their safety certificate ensures long-term safety (forward invariance in probability) in systems with latent variables, (ii) that it can be computed from observed (possibly biased) data despite distribution shifts, (iii) that it can consistently find feasible safe actions in real time, and (iv) that combining causal reinforcement learning with safe control is a novel contribution enabling efficient risk quantification. Overall, these claims are well-supported by the content of the paper, though a few warrant scrutiny:

For Theorem 3.2: The paper provides a rigorous formulation of the safety objective as a probabilistic forward-invariance condition (inequality (6) in the text) and shows that if certain action constraints are satisfied at each step, the probability of remaining in the safe set up to the horizon is at least $1-\epsilon$​. The main condition is never decreasing the safety probability at each timestep (inequality (15)), I am not sure how realistic this is, however given this assumption the proof follows convincingly.

The authors assert that their method can utilize offline data, even if the offline and online transition dynamics differ due to latent factors​. They support this by introducing a procedure (Algorithm 1) to transform the offline dataset into a form ($\tilde D$) that matches the “online” statistical structure needed for their safety analysis​. The claim is plausible given known results in causal inference; indeed, they cite prior works on handling unobserved confounders in offline RL (e.g. using proxy or instrumental variables) to justify that an unbiased Q-function estimate for the true dynamics can be obtained​. While the paper does not deeply detail the statistical assumptions in the main text, it references established causal RL methods (Wang et al. 2021b; Shi et al. 2024) for obtaining those unbiased estimates​. This boosts the credibility of the claim. One potential concern is that the success of this step relies on the offline data being sufficiently rich; the paper assumes the existence of a causal learning solution without extensively discussing failure modes if the data are poor. Nonetheless, given the references and the provided algorithm, the claim appears reasonably supported for scenarios meeting their assumptions (e.g., the latent variables satisfy the stated conditional independence in Assumption 2.1).

The abstract claims the certificate “can continuously find feasible actions” and can be implemented efficiently online​. The theoretical analysis indeed shows that if the system starts in a safe condition, there will always exist some control input that keeps the safety condition satisfied going forward, this is the usual proof technique used in this scenario. As for real-time efficiency, the authors argue that the certificate boils down to evaluating a learned Q-function and checking a constraint inequality at each step, which is computationally light. They do not report actual computation times, but given the form of the controller (essentially a one-step lookahead with a precomputed function), the claim of real-time implementability is believable. This claim would be stronger if accompanied by complexity analysis or timing results; however, no evidence in the paper contradicts it, so it stands as a reasonable assertion.

**Essential References Not Discussed:**

The authors have cited a wide range of relevant work, and no critical references appear to be missing for the topics of interest.

**Experimental Designs Or Analyses:**

The experimental design uses a simulated autonomous driving scenario to evaluate whether the proposed safety certificate performs as expected under latent-variable-induced distribution shift. The scenario is described in enough detail to understand the setup: the observable state $X_t$ seems to include the vehicle’s position and speed (since they mention $X_t^1$ mod 10 and $X_t^2$ which likely correspond to road segment and velocity)​. The latent variable $W_t$ is the road slipperiness, which affects the vehicle’s braking dynamics but is not directly observed by the controller​. The offline dataset is generated by a human driver (behavior policy) that does react to slipperiness (slows down more on slippery roads), although this information is hidden in the data​. This cleverly creates a confounded offline dataset, exactly matching the paper’s problem assumptions. The online controller, by contrast, does not observe $W_t$ and must rely on the learned safety certificate. Beyond my earlier reservations about the scope of the experimental evaluation, the experiments in the paper are well designed and the analysis appears correct.

**Methods And Evaluation Criteria:**

Methods: The methodology is well-suited to the stated problem. The authors formulate the safety assurance problem as maintaining a high probability of safety over an entire horizon (an event of no failures)​, which is appropriate for “long-term” safety considerations. To tackle partial observability and distribution shift, they adopt a confounded MDP framework (a Markov Decision Process with an unobserved variable $W_t$ influencing transitions)​. The core technical approach — deriving safety conditions in probability space — directly addresses the difficulty that latent variables pose for traditional state-space invariance checks​. By using a modified Bellman equation and defining a marginalized value function $V^{\pi}$ and Q-function $Q^{\pi}$ that incorporate the probabilities of staying safe, they convert the safety requirement into conditions on these functions​. This is not a novel idea in and of itself, but it is a smart approach because it allows leveraging the rich toolbox of RL (especially off-policy evaluation) to compute safety-related quantities that would otherwise require knowing the latent dynamics.

The experimental evaluation, while limited in scope, uses appropriate metrics for the safety problem. The primary metrics are the probability of safety at each time step and the long-term safety probability (the probability of no failure up to time $t$. The baseline method chosen for comparison is appropriate: they use a discrete-time control barrier function (DTCBF) method from Cosner et al. 2023 as the benchmark. This baseline represents a state-of-the-art safety filter that does not account for latent-variable-induced distribution shifts (it uses offline estimates naively). Comparing against DTCBF highlights the exact problem the paper is trying to solve. However, I would argue more baseline comparisons  (e.g., a naive RL agent with a reward penalty for failures, or a robust control method that assumes worst-case dynamics) might further strengthen the evaluation. However, given the novelty of the problem setup, I can understand that suitable alternative methods are limited.

The only real limitation in evaluation is the narrow range of experiments – primarily one scenario is tested. Possibly additional scenarios, inlcuding highly-non linear scenarios should be tested.

**Other Comments Or Suggestions:**

Adding a paragraph (perhaps in the conclusion or end of the introduction) explicitly stating the assumptions (like the latent variable independence and the need for sufficient exploration in data) and the current limitations (e.g., requirement for episodic tasks, needing an offline dataset) would increase transparency.

**Other Strengths And Weaknesses:**

Strengths:

Originality: The integration of causal inference techniques with control-theoretic safety guarantees is highly original.
Motivation: Ensuring safety in autonomous systems under real-world uncertainties is extremely important.
Writing: I have no issued with high level writing in the paper it is easy to follow.

Weakness:

Poor scope of the empirical evaluation: As noted, the experiments are limited to one scenario and a relatively simple one at that.

Reliance on accurate Q-function estimation: A potential practical weakness is that the method hinges on obtaining an unbiased estimate of the marginalized Q-function under the true dynamics. The paper assumes this can be done using existing causal RL algorithms​, but in practice these algorithms may require a lot of data or careful tuning. If the Q-function estimate has error, the safety guarantee might not strictly hold (unless the method accounts for estimation error margins, which wasn’t clearly discussed).

Lack of discussion about limitations: The approach might be brittle if the Q-function is imperfect. The paper does not deeply discuss the effects of function approximation or finite-sample errors on the safety guarantee – presumably a small approximation error could violate the guarantee. This is not dealt with in the current submission and represents a potential weakness if one were to implement this in the real world.

Some additional implicity assumptions not discussed in the paper: here are implicit assumptions: e.g., that the latent variables obey Assumption 2.1 (conditional independence and Markov property)​, that the offline data policy had enough randomness (so that one can infer the effect of actions independent of the confounder), and that the model class for Q-function is flexible enough. If any of these assumptions are violated (for example, if the latent variable has temporal correlations, or if the behavior policy is deterministic given $W$), the approach might need modifications.

Technical clarity: While the high-level writing is clear, the theoretical sections (Section 3 in particular) likely involve heavy notation and could be challenging to parse for readers not already familiar with causal MDPs.

**Questions For Authors:**

1. How rich does the offline dataset $D$ need to be to successfully implement the proposed safety certificate? For instance, does $D$ need to cover near-failure scenarios or a wide range of latent variable conditions to learn an accurate Q-function? If the offline data is limited or collected from a narrow policy, how would that impact the safety guarantee?

2 .Could you elaborate on how exactly the unbiased estimate of $Q^{\pi}$ is obtained using causal RL?

3. What happens if the learned $Q^{\pi}$ function has some error? In practice, any data-driven estimate will have uncertainty. Does the framework allow incorporating a confidence bound (e.g., using a slightly smaller threshold to account for estimation error)? If the estimated Q is optimistic by mistake, could the safety guarantee be violated?

4. How computationally intensive is the safety certificate checking and Q-function computation in practice?

5. Your safety objective is defined for a finite horizon $H$. How would the approach extend to an infinite-horizon setting or an ongoing (non-episodic) task?

6. The analysis assumes the latent variable has no memory (i.e., is i.i.d. conditioned on the current state). How would the safety certificate approach handle a latent variable that is stateful or autocorrelated (for example, a hidden system mode that persists over time)? Would the method require an extension to POMDP belief-state tracking, or is there a way to incorporate such correlations in the current framework?

**Relation To Broader Scientific Literature:**

The paper positions itself at the intersection of safe control (especially forward invariance and barrier functions) and causal reinforcement learning (offline RL with unobserved confounders). The authors do a good job discussing relevant prior work in both areas and highlighting the gap that this paper fills.

On the safe control side, they cite classic and recent works on Lyapunov/barrier functions and safety filters for both deterministic and stochastic systems​. They acknowledge methods dealing with partial observation via robust or belief-space barrier functions​. Importantly, they point out that these existing methods generally assume either known dynamics or at least no systematic bias between the data used for design and the true deployment scenario​.

Overall, the paper’s positioning in literature is well described. It properly credits a wide array of relevant prior work from both the control theory community and the RL/causal community. The combination of these threads itself highlights the novelty: none of the cited works alone addresses what the paper does at their intersection. The authors make it clear that stochastic safe control and causal RL have “been studied in isolation until now”.

**Theoretical Claims:**

The theoretical claims in the paper are logically sound and align with known principles in safe control and Markov decision processes. The central theoretical result is that if a certain safety certificate condition is satisfied for all times, then the safety objective (the probabilistic invariance condition (6)) is guaranteed​. This claim is very much in line with the concept of forward invariance: if one can ensure a condition holds at each step (and that condition is designed to enforce future safety), then by induction the system remains safe over the horizon.

The modified Bellman equation and the link between the risk measure and the marginalized Q-function are credible and have been explored in different settings before.

---

> ### Author Rebuttal · Authors · 2025-04-01
>
> We sincerely appreciate your efforts and valuable feedback. Please find below our answers to your questions.
>
> Q1, Q2.
> The required offline dataset and the estimation procedures for $Q^{\pi}$ depend on the specific choice of causal RL techniques. For example, the Q-estimator in Shi et al. 2024 is persistent, and the authors demonstrated in simulation that the mean squared error decreases at the scale of Q(exp(1/MSE)). The techniques in Wang et al. 2021 are shown to have $\sqrt{T}$ regret.
> In general, when no model information is available, sufficient data is needed to gather information about the transition dynamics of all states and to learn the Q-function. When some physics information is available, the physics information can be used to reduce the requirement on data points around failure scenarios (such as in Hoshino & Nakahira, 2024).
> We also would like to mention that the proposed framework and its theoretical results---Proposition 3.1 (conversion of long-term safe probability into value-function), Theorem 3.2 (safety condition), and Theorem 3.2 (persistent feasibility)--- hold irrespective of the specific choice of causal RL methods. In the paper, we have demonstrated how to use the RL method of paper 2, but one can replace line 14 in Algorithm 2 (which uses Shi et al. 2024) with other Q-estimators. Data-efficient causal RL methods to estimate $Q^{\pi}$ are beyond the scope of this paper: Rather, we focus on how to design a persistently feasible safety certificate that can leverage external causal RL methods.
>
> Paper 1: Shi, Chengchun, et al. "Off-policy confidence interval estimation with confounded markov decision process." Journal of the American Statistical Association 119.545 (2024): 273-284.
>
> Paper 2: Wang, Lingxiao, Zhuoran Yang, and Zhaoran Wang. "Provably efficient causal reinforcement learning with confounded observational data." Advances in Neural Information Processing Systems 34 (2021): 21164-21175.
>
> Q3.
> Our future work is to account for the errors in the estimated $Q^{\pi}$ function. One approach is to compute a confidence interval around the estimated Q-function and adjust the safety threshold downward to guard against potential optimism. This would involve characterizing or bounding the errors arising from distributional mismatch and accounting for such errors in the derivation of the safety condition, which would help ensure that the safety guarantee is maintained even when estimation errors occur.
>
> Q4.
> The online evaluation reduces to finding a control action that satisfies the safety condition (Eq. (25)). For example, the evaluation can be done by solving an optimization problem in Eq. (43), or projecting a nominal control action $U^n$ on the constraint set $S(X_t,u,t)\geq 0$. The function $S(X_t,u,t)$ can be computed offline and stored (e.g., in a neural network), and the control action $u$ is the only optimization variable. We expect the proposed method to not require significant computation online.
>
> Q5.
> While extension to an infinite time horizon or a non-episodic task is beyond the scope of this paper, the high-level approach of our method is likely to admit an extension to such settings. For such settings, one can first find an auxiliary stochastic process based on observed statistics that can be used to explicitly differentiate online vs. offline statistics (like Eqs. (8) and (11)). One can explore different definitions of safety and adjusted value/Q-functions that still allow for relating long-term safety with adjusted value/Q-functions and for obtaining a safety condition based on such functions, e.g., incorporating a discount factor for future risk.
>
> Q6.
> The current setting (confounded MDP) is a special case of POMDP where one can use the Markov nature of the observed state to design fast algorithms for a safety certificate. For latent variables that exhibit finite memory or autocorrelation, one can augment the state space with the necessary memory such that the augmented state is Markov. For most general POMDPs without any additional structures (which is by itself a hard problem), belief state is an option. We expect that, for belief state, the high level ideas to construct an auxiliary stochastic process based on observed statistics to explicitly differentiate online vs offline statistics (like equation (8) and (11)), approaches to leverage existing Q-estimator to obtain long-term safety, and constructing safety certificate based on Q-functions, is likely to still hold.

---

### Official Review · Reviewer_GFiT · 2025-03-16

**Overall Recommendation:** 4

**Summary:**

This paper proposes a probabilistic safety certificate design methodology for systems with latent variables, aiming to address the safety verification challenges faced by traditional control methods in scenarios involving partial observability and distribution shifts. The core framework involves:
1. **Specialized MDP Formulation**: A tailored Markov Decision Process (MDP) is defined for both offline and online settings, where the value function of this MDP is proven to correspond to the safety probability.
2. **Safety Certificate via Value Function**: The safety certificate is derived from the value function, ensuring that adherence to this certificate guarantees long-term safety.
3. **Q-Function-Based Computation**: Due to the computational infeasibility of directly calculating the expected value function under online data, it is demonstrated that the safety certificate can alternatively be computed using a specific Q-function. The optimal policy derived from this Q-function inherently satisfies long-term safety constraints.
4. **Causal Reinforcement Learning Integration**: The paper further proves that the required Q-function can be efficiently estimated through causal reinforcement learning techniques. By leveraging this Q-function to adjust control actions, the methodology ensures that every executed action maintains safety guarantees.

This approach bridges safety-critical control and causal inference, providing a theoretically grounded and computationally tractable solution for systems operating under latent variables and distribution shifts.

**Claims And Evidence:**

Equation 42 is presented in a discrete form. Does this imply that the proposed method is only applicable to discrete action spaces? How can the approach be extended to continuous action spaces?

**Essential References Not Discussed:**

In the paper 'Safe Reinforcement Learning by Imagining the Near Future' (https://arxiv.org/abs/2202.07789), Equation 2 also demonstrates that modifying the MDP structure enables the Q-function to be used for safety evaluation.

**Experimental Designs Or Analyses:**

1. The manuscript introduces the estimators $\hat{P}_U(U_t | \hat{Y}_t) $ and $ \hat{P}_M(M_t | \hat{Y}_t, U_t) $, but their practical interpretation remains unclear. Could the authors provide a concrete explanation of these terms using the example from Section 4 (Numerical Simulation)?

2. Equation 25 is presented without sufficient detail on its computation. Could the authors elaborate on how this equation is calculated, using the specific context of the numerical example in Section 4? A step-by-step explanation would greatly enhance the reproducibility of the results.

3. The role of the mediator $ M_t $ is not clearly defined in the context of the proposed method. Could the authors clarify what $M_t$represents in the numerical example from Section 4? For instance, is it an observed variable, a latent variable, or a function of other system states? A detailed explanation would help readers better understand its significance in the framework.

**Methods And Evaluation Criteria:**

The paper relies on two critical assumptions:
1. The existence of a consistent estimator $\ \hat{P}_M(M_t | \hat{Y}_t, U_t)$ for the distribution $P(M_t | X_t, U_t) $.
2. The existence of a consistent estimator $\hat{P}_U(U_t | \hat{Y}_t) $ based on the offline dataset $\tilde{D} $, which estimates the empirical distribution of the control action $U_t $ given the observable state $\hat{Y}_t $.

While these estimators are assumed to converge to the true distribution parameters as the sample size approaches infinity, practical limitations arise when the dataset is finite. In such cases, the state transition dynamics in the offline dataset may not align with those in the online environment.

In most real-world scenarios, these assumptions are unlikely to hold. For instance, in autonomous driving, it is impossible to collect exhaustive human driving data or fully capture the environmental transition distributions. The key concern is: What happens when these assumptions are violated? The paper does not adequately address the robustness of the proposed method under such conditions, nor does it provide mitigation strategies for scenarios where the dataset is insufficient or non-representative."

**Other Comments Or Suggestions:**

Minor Correction: In line 181, the sequence $ \bar{X}_{0:H} $ is described as having the 'online statistics '.this should be corrected to 'offline statistics,' as the context suggests the use of pre-collected data rather than real-time data.

**Other Strengths And Weaknesses:**

I have carefully reviewed the majority of the mathematical proofs in this paper and found no significant issues. The theoretical foundations presented by the authors are rigorous and well-constructed.

**Questions For Authors:**

The paper states: 'The nominal policy simply chooses actions in the action space with identical probability.' Could the authors clarify the role of the nominal policy here?

**Relation To Broader Scientific Literature:**

It could potentially be applied to imitation learning and other systems to learn safe policies.

**Theoretical Claims:**

1. The derivation of Equation 42 is not sufficiently detailed in the manuscript. Could the authors provide a step-by-step derivation to clarify how this equation is obtained? Additionally, the origin of the optimal action $ u^* $ is unclear. If $ u^* $ is part of the action space $u $, how is it specifically derived or selected?

---

> ### Author Rebuttal · Authors · 2025-04-01
>
> We sincerely appreciate your efforts and valuable feedback. Please find below our answers to your questions.
>
> Question regarding discrete action:
> Equation (42) and its neighbors explain the technique of Shi et al. 2024. N can be used to estimate $Q^\pi$ in our problem. While this specific method is developed primarily for a discrete action space, the proposed framework is not restricted to any specific choice of confounded MDP technique. Our main theoretical results—Proposition 3.1 (relation between long-term safety and value-function), Theorem 3.2 (safety condition), and Theorem 3.2 (persistent feasibility) do not require discrete action spaces. Thus, other techniques to learn the Q-function in a continuous action space, combined with the above theoretical results, can be used to handle a continuous action space.
>
> Regarding the impact of the latent variable and limited data:
> Unobserved variables and limited data are two different factors that contribute to errors in Q-learning. Without proper adjustment, even with infinitely many data, the learned Q function or quantified risk will not converge to the true values due to the distribution shift arising from latent variables: this is the problem we study in this paper. This issue has not been studied in the context of safety certificates, and the proposed method is expected to have less distribution mismatch (due to the adjustment for latent variables) than existing approaches that treat the observed distribution as the true dynamics.
>
> This issue of uncertain systems with limited data is another problem that the existing tools in the safety certificate cannot handle well. Many existing tools for safety certificates require knowledge of accurate system dynamics and ground truth distributions. When the system is uncertain and needs to be inferred from limited data, existing methods, ranging from system identification to value estimation to reachability analysis, would all face this problem. In our approach, this problem appears in the form of a potential distribution mismatch in the Q-function. In other stochastic safe control approaches (which typically require a characterization of the probability of forward invariance), this problem appears in the form of errors. Regarding the influence of limited data volume versus the influence of latent variables, we don’t have a quantitative sense of which one is more significant, as this is not within the scope of the paper. We agree with the reviewer that a natural and interesting next step is to account for this. We will add this aspect to the discussion.
> ‘Essential References Not Discussed’: Thanks for pointing out this missing reference for us. We obtained the idea of modifying the MDP structure from https://arxiv.org/abs/2403.16391, so we cited this paper as the source. As the paper you pointed out has this idea and is earlier, we will cite it as well.
>
> Response to ‘Questions For Authors’: The nominal policy is the policy with respect to which the long-term safe probability is evaluated, i.e., the $\pi$ in the long-term safe probability defined in (5).
>
> Question in ‘Supplementary Material’: What contributes to the discrepancy most is the behavioral policy. Consider that the behavioral policy knows that when the latent variable W is 1, choosing action 1 will lead to an unsafe state. It will then avoid choosing 1, but will only avoid that when W is 1. When we sample from this behavioral policy, we never know the consequence of choosing action 1 because the behavioral policy avoids doing so when it is risky, regardless of how many samples we have. We don’t know when choosing action 1 is risky because we cannot observe the latent variable. In fact, if action 1 is completely safe when W is 0, we will have a dataset that the next state is always safe when action 1 is chosen.
>
> Questions in ‘Claims and Evidence’, ‘Theoretical Claims’, ‘Experimental Design and Analyses’ 1 and 3: These questions are all about Section 3.4, so we answer them altogether. The entire Section 3.4 considers an example algorithm using the Q-function estimation method proposed in https://arxiv.org/abs/2202.10589. The assumptions about the discrete action space, the existence of the mediator and the existence of the unbiased estimator are all from that paper. The proposed method itself does not require these assumptions if the Q-function estimation method does not require them. Regarding the derivation of (42), please check Section 3.6 of the aforementioned paper, as (42) is an equation from that section.
>
> Question in ‘Experimental Design and Analyses’, 2: Since we have a Q-function estimation, the first term is given. If the action space is discrete, one can compute the second term using the given policy and its associated distribution of actions. If the action space is continuous, then the expectation is given in the form of an integral, which can be approximated using various methods including Monte-Carlo.

---

> > ### Comment · Reviewer_GFiT · 2025-04-05
> >
> > I have read the authors’ response. However, I still find that the authors did not clearly address the three main concerns I raised in the “Experimental Designs Or Analyses” section.

---

> > > ### Author Response · Authors · 2025-04-08
> > >
> > > Thank you very much for carefully reviewing our response. We apologize for not providing a detailed answer to your questions in “Experimental Designs or Analyses” due to the character limit. Please find below for more details. As some of the earlier questions inform the responses to later ones, we organize our answers in the order of Q3, Q1, and then Q2. For notational simplicity, we use $X$ to denote the observable states, but $X$ can also be other variants of the observable state (e.g., one can convert $X$ into $\hat{X}$ and $\hat{Y}$ using line 5-12 of Algorithm 1).
> > >
> > > Role of the mediator variable $M_t$ (Q3):
> > >
> > > The mediator $M_t$ is an observed variable satisfying Assumption 3.5 that helps address distribution shifts between offline data and online statistics of the observed variables ($P_{offline}(X_{t+1}|X_t,U_t) \neq P_{online}(X_{t+1}|X_t,U_t)$). Specifically, one essential part in the proposed method is to characterize $P_{online}(X_{t+1}|X_t,U_t)$, which requires identifying the impact of current action on future state conditioned on current state (i.e. $U_t \rightarrow X_{t+1}|X_t$). Since the direct path $U_t\rightarrow X_{t+1}|X_t$ in the offline data is confounded by another spurious path involving the latent variable $U_t \leftarrow W_t \rightarrow X_{t+1}|X_t$, we split $U_t \rightarrow X_{t+1}|X_t$ into two paths: $U_t\rightarrow M_t|X_t$ and $M_t\rightarrow X_{t+1}|X_t$. Here, $M_t$ can be interpreted as an intermediate variable used in the decomposition (condition 1 of Assumption 3.5) such that the online distribution of each path is either known or can be computed using offline data (conditions 2 and 3 of Assumption 3.5).
> > >
> > > Accordingly, the mediator variable is assumed to satisfy three conditions in Assumption 3.5. 1. $M_t$ intercepts every directed path from the control action $U_t$ to $X_{t+1}$, ensuring the problem can be decomposed into the two paths. 2. $X_t$ blocks all backdoor paths from $U_t$ to $M_t$, which ensures the distribution $M_t|U_t,X_t$ does not exhibit distribution shifts due to latent confounding. 3. all backdoor paths from $M_t$ to $X_{t+1}$ are blocked by $(X_t,U_t)$. These three conditions jointly allow $V^l_{\gamma}$ in (42) to be computed from the offline statistics. This is borrowed from Shi et al. 2024 and Wang et al. 2021 and is built based on the front-door adjustment formula in causal inference (Chapter 3.3.2 of Pearl. 2009).
> > >
> > > Consider driving on a potentially slippery road (case study used in Sec. 4). The latent variable $W_t$ is the road slipperiness, which is not observable by the safety certificate online. The offline dataset is generated by a human driver (behavior policy) who may react to slipperiness. The control action $U_t$ is the intended brake/throttle, and the mediator $M_t(=U_t+N^1_t)$ is a measurable, effective brake/throttle level.
> > >
> > > Practical interpretation of the estimators (Q1):
> > >
> > > The estimator $\hat{P}_U(U_t|\hat{Y}_t)$ estimates the conditional distribution $U_t|\hat{Y}_t$ marginalized over $W_t$ in the offline data. Note that this is not the behavioral policy ($\pi^b(U_t|X_t,W_t)$), but $p_a^*$ in Shi et al. 2024. The estimator $\hat{P}_M(M_t|\hat{Y}_t,U_t)$ estimates the empirical distribution of the mediator given the observable state and action. In the above driving setting, $\hat{P}_U(U_t|\hat{Y}_t)$ estimates the empirical (offline) distribution of the intended brake/throttle given the location and velocity of the vehicle. The estimator $\hat{P}_M(M_t|\hat{Y}_t,U_t)$ estimates the distribution of the effective brake/throttle given the intended brake/throttle and observed state.
> > >
> > > Computation of (25) (Q2):
> > >
> > > The computation of (25) requires the Q-function $Q^\pi$ and the control policy $\pi$. The control policy is given since this is the policy we would like to evaluate the long-term safety with respect to, as defined in (7). In Algorithm 2, $Q^\pi_\gamma$ is first estimated in lines 5-8 through the existing method in Shi et al. 2024. In line 14, $Q^\pi$ is computed using (40) with $Q^\pi_\gamma$ and consistent estimator $\hat{P}_M(M_t|\hat{Y}_t,U_t)$. In line 15, (43) is computed, which requires computing (25). The expectations in (25) and (40) can be computed using sampling-based methods or direct computation using the distribution over which the expectation is taken. In the simulation, unbiased estimation of $Q^\pi$ is assumed to be given, so the only task is to compute (25) using $Q^\pi$, where we use Monte-Carlo method to estimate the expectation in (25). In the simulation, we use $\pi=\pi^n$, which is given by a policy that selects control actions with identical probabilities.
> > >
> > > We are approaching the character limit. If you have any further questions, please feel free to submit an additional response—we would be more than happy to provide further details.

---

### Decision · Program_Chairs · 2025-05-01

**Decision:**

Accept (poster)

**Comment:**

The authors propose a probabilistic safety certificate design methodology for systems with latent variables. They have a specialized MDP formulation and the safety certificate is derived from the value function. They use Q-function computation and show that it can be efficiently estimated through causal reinforcement learning techniques.

The reviewers found the contributions of this paper interesting. I recommend the authors to address the the reviewers' remaining concerns in the final version.